# Chromosome segregation fidelity requires microtubule polyglutamylation by the cancer downregulated enzyme TTLL11

Ivan Zadra[1], Senda Jimenez-Delgado[1,6], Miquel Anglada-Girotto [1,6], Carolina Segura-Morales [1], Zachary J. Compton[1], Carsten Janke [2,3], Luis Serrano [1,4,5], Verena Ruprecht[1,4,5] & Isabelle Vernos [1,4,5] ✉

Regulation of microtubule (MT) dynamics is key for mitotic spindle assembly and faithful chromosome segregation. Here we show that polyglutamylation, a still understudied post-translational modification of spindle MTs, is essential to define their dynamics within the range required for error-free chromosome segregation. We identify TTLL11 as an enzyme driving MT polyglutamylation in mitosis and show that reducing TTLL11 levels in human cells or zebrafish embryos compromises chromosome segregation fidelity and impairs early embryonic development. Our data reveal a mechanism to ensure genome stability in normal cells that is compromised in cancer cells that systematically downregulate TTLL11. Our data suggest a direct link between MT dynamics regulation, MT polyglutamylation and two salient features of tumour cells, aneuploidy and chromosome instability (CIN).

Chromosome segregation fidelity is essential for genome integrity and viability of cells and organisms. Errors can generate aneuploidy, an imbalance of chromosome number per cell, one of the most common causes of miscarriage[1] and a hallmark of cancer[2]. The spindle is the molecular machine that provides the support and force required for chromosome segregation. Its transient assembly and function rely on the fine regulation of its main constituents, the microtubules (MTs). While MT dynamics increase globally at the transition from interphase to mitosis[3], spindle organization and function involve a local partial stabilization of MTs around the chromosomes. As the spindle assembles, some MTs establish reversible attachments with the chromosomes via their kinetochores and form kinetochore fibers (K-fibers) that align the chromosomes to the metaphase plate and then pull the sister chromatids to the spindle poles during anaphase. The stochastic nature of MT capture by the kinetochores in early mitosis generates a large number of erroneous attachments that need to be corrected before the onset of anaphase. This critical process is monitored by the spindle assembly checkpoint (SAC), a surveillance mechanism that delays anaphase until all chromosomes are attached to MTs that connect them to the opposite spindle poles[4], a pre-requisite for faithful chromosome segregation. Although the SAC senses most incorrect kinetochore–MT attachments, it fails to sense merotelic attachments occurring when a single kinetochore is simultaneously attached to both spindle poles, cells can therefore enter anaphase generating lagging chromosomes that often result in aneuploidy[5].

In the initial phases of mitosis, MTs need to maintain stable enough interactions with kinetochores to move and align the chromosomes to the metaphase plate, yet remain dynamic enough to allow for error correction. While major changes (increases or decreases) in MT dynamics preclude spindle assembly and chromosome segregation altogether, more subtle changes directly impact chromosome segregation fidelity[6]. Previous studies showed that the partial stabilization of MTs in mitosis[7] leads to an increased rate of chromosome mis-segregation[8], indicating that a fine regulation of MT dynamics within a narrow range is essential to ensure chromosome segregation fidelity[9]. Although no clear genetic signature (mutations or deletions)

[1]Centre for Genomic Regulation (CRG), The Barcelona Institute of Science and Technology, Barcelona, Spain. [2]Institut Curie, Université PSL, CNRS UMR3348, F-91401 Orsay, France. [3]Université Paris-Saclay, CNRS UMR3348, F-91401 Orsay, France. [4]Universitat Pompeu Fabra (UPF), Barcelona, Spain. [5]ICREA, Pg. Lluis Companys 23, Barcelona 08010, Spain. [6]These authors contributed equally: Senda Jimenez-Delgado, Miquel Anglada-Girotto. ✉e-mail: isabelle.vernos@crg.eu

or molecular biomarkers have been identified so far, several pieces of data link changes in the dynamics of kinetochore or spindle MTs to aneuploidy and CIN in cancer cells. For instance, cancer cells with CIN have hyperstable kinetochore–MT (K-MT) attachments when compared to chromosomally stable diploid cells[6,9]. Moreover, an increased K–MT half-life in cancer cells appears to be sufficient to induce CIN[6] and an increased MT assembly rate correlates with CIN in colorectal cancer cells[10]. Understanding how MT dynamics is finely regulated during mitosis is therefore key to understanding the basis for chromosome segregation fidelity and may provide novel insights into aneuploidy and CIN in cancer cells.

Tubulin post-translational modifications (PTMs) are regulators of MT function in differentiated cells. During mitosis, spindle MTs are modified by numerous PTMs, including acetylation, detyrosination and polyglutamylation[11,12] but relatively still little is known about their roles in cell division. It was recently shown that detyrosination guides metaphase chromosome congression via the motor protein CENP-E[13] and plays a role in error correction and mitotic fidelity in human cells[14]. In addition, acetylation was recently reported to play a role in the maintenance of spindle bipolarity in epithelial cells[15]. Here we identify TTLL11 (tubulin tyrosine ligase like 11) as an enzyme that specifically polyglutamylates the spindle MTs and we address the role of TTLL11 and spindle MT polyglutamylation during mitosis, in HeLa cells and zebrafish embryos. Our results show that TTLL11-dependent MT polyglutamylation provides a central mechanism that finely controls MT dynamics in the mitotic spindle that may explain why chromosomes rarely mis-segregate in normal tissues[16]. This mechanism is at stake in human cancers through the systematic downregulation of TTLL11.

## Results

### TTLL11 polyglutamylates spindle MTs

To address the functional implications of polyglutamylation of spindle MTs, we first aimed to identify the enzyme(s) driving this PTM during mitosis. Since there are no available antibodies to monitor the endogenous localization of the eight human TTLL glutamylase enzymes[17], we expressed each one in HeLa cells as a recombinant protein in two forms: with a fluorescent tag at either its C-terminus or its N-terminus. We then checked the localization of the recombinant enzymes by immunofluorescence in mitotic cells (Supplementary Fig. 1a). Two of these enzymes, TTLL11 and TTLL13, localized to the spindle (irrespective of the position of the fluorescent tag), whereas none of the others showed any specific localization in mitotic cells (Supplementary Fig. 1a). We then examined the expression patterns of TTLL11 and TTLL13 in human tissues (Supplementary Fig. 1b). While TTLL13 was poorly expressed in general, TTLL11 was widely expressed in the majority of human tissues. Similar results were found for the most common cell lines used in research (see https://www.proteinatlas.org/). Consistently, TTLL11 but not TTLL13 was expressed in HeLa cells (Supplementary Fig. 1c). These data pointed to TTLL11 as a good candidate for driving tubulin polyglutamylation in mitosis. To test this hypothesis, we silenced TTLL11 expression with siRNAs in HeLa cells (Supplementary Fig. 2a, b) and quantified the level of MT polyglutamylation in the spindle by immunofluorescence using: i) the monoclonal antibody GT335, which is specific for the branching point of glutamate side chains[18], and ii) the polyclonal antibody PolyE, which only detects glutamate chains of three or more residues[17] (Supplementary Fig. 2c). The GT335 signal in interphase and mitosis was very similar in control and siTTLL11 cells in both intensity and distribution (Supplementary Fig. 2f, g). The low PolyE signal in interphase cells was also similar for control and siTTLL11 cells (Supplementary Fig. 2d, e). In contrast, the PolyE signal was specifically and significantly reduced in spindles assembled in siTTLL11 cells as compared to control cells ($0.52 \pm 0.15$ a.u. versus $0.82 \pm 0.19$ a.u., respectively; $P < 0.0001$) (Fig. 1a, b). Consistently, TTLL11 silencing also reduced spindle MT polyglutamylation in the non-transformed human cell line hTERT-RPE1 (Supplementary Fig. 3a, b). The reduction of MT

polyglutamylation levels was specific and no changes in the levels of other PTMs (tyrosination, detyrosination, or acetylation) were detected in siTTLL11 spindles (Fig. 1c). Altogether, these data showed that elongation of glutamate chains on the spindle MTs but not branching was abrogated in the TTLL11-silenced cells. This was consistent with previous studies that showed that TTLL11 is a polyglutamylase with a preference for side-chain elongation[17]. Altogether, our results point to TTLL11 as an enzyme that generates long glutamate chains specifically on the spindle MTs.

### TTLL11 is essential for chromosome segregation fidelity

To investigate the functional consequences of MT polyglutamylation on spindle assembly and chromosome segregation, we used time-lapse imaging to monitor TTLL11-silenced HeLa cells undergoing mitosis. Both TTLL11-silenced and control cells assembled bipolar spindles (Supplementary Movie 1) and aligned and segregated chromosomes with a similar timing. Consistently, their mitotic index was also comparable (Fig. 1d). Strikingly, however, the fidelity of chromosome segregation during anaphase was compromised in TTLL11-silenced HeLa cells which showed a significant increase in chromosome segregation defects, including lagging chromosomes, as compared to control cells ($25.5\% \pm 5.9\%$ versus $8.6\% \pm 2.2\%$, respectively; $P = 0.001$) (Fig. 1e). Chromosome segregation fidelity was also compromised in siTTLL11 hTERT-RPE1 cells (Supplementary Fig. 3c).

To confirm these results in vivo, we used zebrafish embryos as a model system. Immunofluorescence analysis using the PolyE antibody showed that spindle MTs are polyglutamylated in blastula-stage embryos (e.g., at 4 h post fertilization (hpf)) (Fig. 2a). In addition, RT-PCR showed that the highly-conserved TTLL11 zebrafish ortholog (Supplementary Fig. 4a, b) is expressed in early embryos from the 4-cell stage until 24 hpf (Fig. 2b). Time-lapse imaging of 4-hpf blastula embryos expressing GFP-zebrafish (zf)TTLL11 showed that the fluorescent protein localized to the spindle, as observed in HeLa cells (Fig. 2c and Supplementary Movie 2) and sometimes also transiently to the chromosomes although not consistently.

To address the role of MT polyglutamylation by TTLL11 in zebrafish embryo development, we used morpholinos (MO) to downregulate zfTTLL11 and scored the phenotype of the resulting embryos. In contrast to control-MO–injected embryos, embryos injected with zfTTLL11-morpholino (zfTTLL11-MO) had increased levels of early lethality ($44.9\%$ versus $2.6\%$ for controls), and $54.5\%$ showed severe developmental defects at 36 hpf (Fig. 2d, e and Supplementary Table 1). To rule out any potential off-target effects of the zfTTLL11-MO, we co-injected embryos with zfTTLL11-MO and an MO-resistant mRNA encoding zfTTLL11 and monitored early embryonic development as before. We found that exogenous expression of zfTTLL11 in zfTTLL11-MO-injected embryos reduced early embryonic death ($25\%$) and also rescued early embryo development at 32 hpf, with an increased percentage of phenotypically normal embryos and a reduced percentage of embryos with severe developmental defects (Fig. 2d, e). The specificity of the MO was further confirmed in F0 biallelic zfTTLL11 knockout embryos that showed embryonic developmental defects similar to the morphants (see "Methods" and Supplementary Fig. 5a, b).

A key residue for the catalytic activity of the TTLL enzyme family is conserved in zfTTLL11 (E466)[17] (Supplementary Fig. 4b). We therefore directly tested whether TTLL11 catalytic activity is required for rescuing early embryonic development of morphant embryos by co-injecting mRNA encoding catalytically dead zfTTLL11 (E466G) with zfTTLL11-MO. Expression of zfTTLL11(E466G) did not rescue early embryonic development of the zfTTLL11-MO embryos. Indeed, these embryos showed major developmental defects comparable to those observed in embryos only injected with zfTTLL11-MO (Fig. 2e). These data showed the catalytic activity of zfTTLL11 is fundamental for zebrafish embryo early development. Altogether, our data suggested that TTLL11 plays an essential role during the early embryonic development phases.

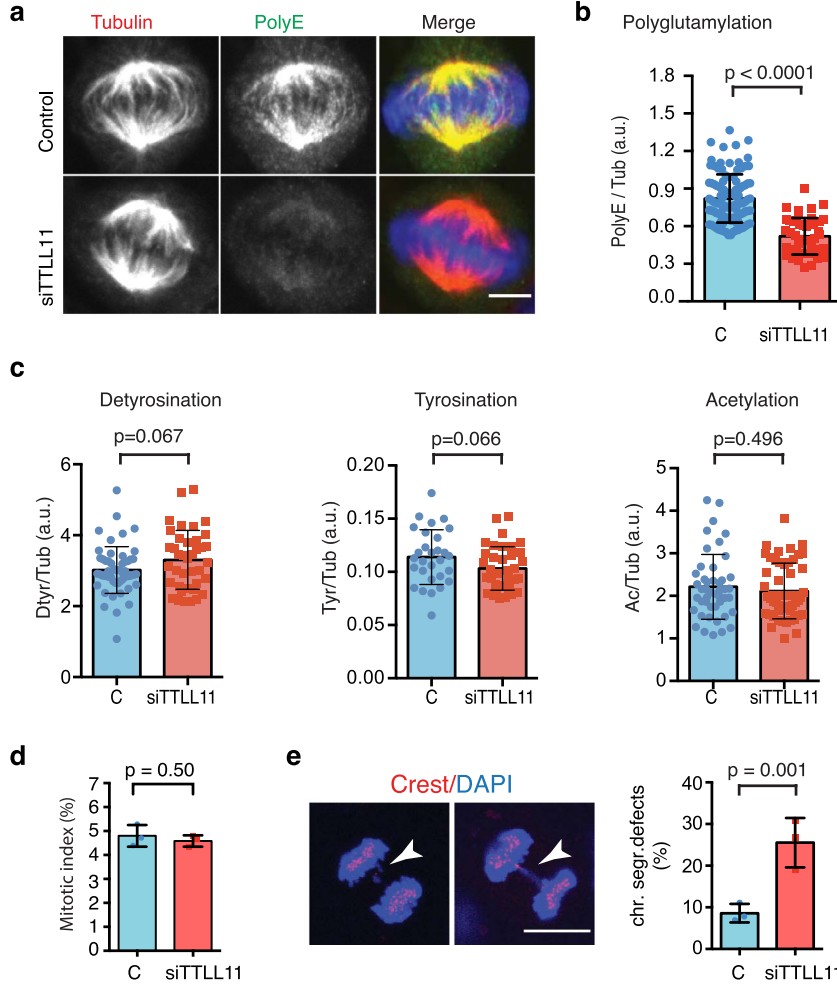

**Fig. 1 | TTLL11 localizes to the spindle and drives MT polyglutamylation in mitosis. a** Immunofluorescence images of metaphase spindles in control and siTTLL11 HeLa cells, showing PolyE (green), tubulin (red) and DNA (blue). Scale bars, 10 μm. **b** Quantification of the polyE signal normalized to the total tubulin signal in control and siTTLL11 spindles. *n* (control) = 105 cells and *n* (siTTLL11) = 50 cells. Graph representative of *N* = 5 independent experiments. Data are presented as mean values +/− SD. **c** Quantification of immunofluorescence signal for detyrosinated, tyrosinated and acetylated tubulin normalized to the total tubulin signal in control and siTTLL11 spindles. Detyrosination: *n* (control) = 55 cells and *n* (siTTLL11) = 39 cells; Tyrosination: *n* (control) = 29 cells and *n* (siTTLL11) = 37 cells. Acetylation: *n* (control) = 37 cells and *n* (siTTLL11) = 37 cells. Graphs representative

of *N* = 3 independent experiments. Data are presented as mean values +/− SD. **d** Mitotic index in control and siTTLL11 cells. *n* = 1000 cells per condition; *N* = 3. **e** Immunofluorescence images of two siTTLL11 anaphase cells showing chromosome segregation defects including a lagging chromosome (white arrows). Kinetochores were labeled with CREST (red) and DNA with DAPI (blue). Quantification of lagging chromosome frequency in control and siTTLL11 anaphase cells. *n* (control) = 349 and (siTTLL11) = 270 cells examined over *N* = 3 independent experiments. Scale bar, 15 μm. Data are presented as mean values +/− SD. All *P* values are based on unpaired two-sided *t* test with 95% confidence. Source data are provided as a Source Data file.

We then directly explored the role of zfTTLL11 in chromosome segregation fidelity during the early phases of zebrafish embryonic development that consist of multiple rounds of successive cell divisions. We first checked the levels of MT polyglutamylation in spindles assembled in control and morphant embryos by immunofluorescence on disaggregated embryos cells with the PolyE antibody. Quantification of the PolyE signal showed that indeed the polyglutamylation of the spindle MTs were significantly reduced in morphant embryos compared to controls (Fig. 2f, g). We then monitored cell division and chromosome segregation in live blastula-stage embryos by 4D time-lapse imaging (Fig. 3). In contrast to the error-free anaphases observed in control embryos, we found that the majority of zfTTLL11-MO-injected embryos (e.g., 16 of 18 embryos) had one or more cells undergoing anaphase with chromosome segregation defects including lagging chromosomes that often formed micronuclei during the observation period in a 40–60 μm stack volume (Fig. 3 and Supplementary Movie 3). These chromosome segregation defects observed in the morphant embryos were reminiscent of those we observed in TTLL11-silenced

HeLa cells. Of note, the quantification of the chromosome segregation defects, including lagging chromosomes in fixed embryos, confirmed these observations: lagging chromosomes were, overall, rare in control embryos (7%; *n* = 41 cells), whereas almost half (49.1%) of anaphase cells in morphant embryos had at least one lagging chromosome (*n* = 46 cells) (Fig. 3a, b and Supplementary Movie 4).

To further rule out any potential off-target effects of the morpholinos, we used a CRISPR/Cas9 approach to generate zebrafish F0 biallelic knockout embryos[19]. Consistently, we also found that 58.9% of these embryos showed developmental defects at 36hpf (42.5% mild and 16.4% severe), (Supplementary Fig. 5c, d).

Both morphant and the F0 biallelic knockout embryos contained micronuclei (Fig. 3c–e and Supplementary Fig. 5e). Micronuclei usually arise from lagging chromosomes (or large chromosomal fragments) that fail to be included in the daughter nuclei at the end of mitosis[20,21]. Indeed, live imaging of dividing cells in morphants showed that it was possible to trace the origin of some micronuclei to a lagging chromosome (or large chromosome fragment) (Fig. 3c and Supplementary

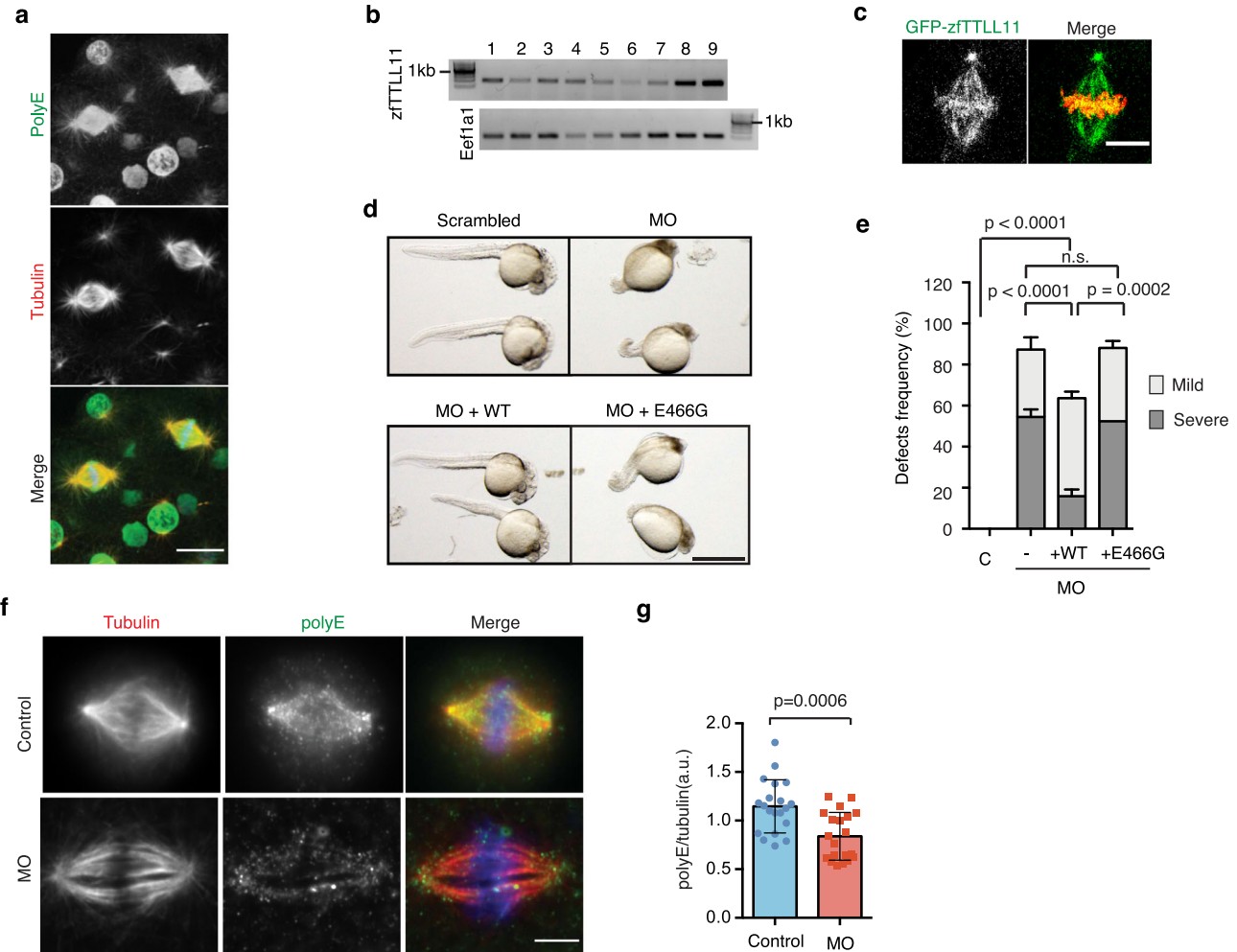

**Fig. 2 | zfTTLL11 is required for early embryonic zebrafish development.**
**a** Immunofluorescence image of a zebrafish embryo (at 4 h post fertilization [hpf]), showing PolyE (green), tubulin (red). Scale bar, 20 μm. Representative image of N = 3 independent experiments. **b** Semiquantitative RT-PCR showing zfTTLL11 expression in zebrafish embryos. Lanes 1, 4-cell stage; 2, 8-cell stage; 3, 64- cell stage; 4, 256-cell stage; 5, sphere; 6, shield; 7, 70% epiboly; 8, 90% epiboly; and 9, 24 hpf. Eef1a1 was amplified as a control. Representative blot of N = 4 independent experiments. **c** Fluorescent image of a spindle in a zebrafish embryo expressing GFP-zfTTLL11 (green) and H2B-mCherry (red). Scale bar, 10 μm. Representative image of N = 3 independent experiments. **d** Zebrafish embryos (36 hpf) injected at the zygote stage with scrambled MO (control), zfTTLL11-Morpholinos (MO), MO and zfTTLL11 mRNA (WT) or MO and catalytically inactive zfTTLL11 (MO + E466G)

mRNA. Scale bar, 1 mm. **e** Cumulative bar plot of developmental defects (severe or mild) in 36-hpf embryos of N = 5 independent experiments, representative of a total of four independent experiments (≥20 embryos scored per condition). ***P < 0.001; ****P < 0.0001; P values are based on a two-tailed $\chi^2$ test with a 95% confidence interval. Error bars represent SD. **f** Immunofluorescence images of metaphase spindles from dissected cells from control and MO zebrafish embryos, showing PolyE (green), tubulin (red), and DNA (blue). Scale bar, 5 μm. **g** Graph showing the quantification of the polyE signal normalized to the total tubulin signal in control and siTTLL11 spindles. n (control) = 28 cells and n (siTTLL11) = 26 cells. Graph representative of N = 2 independent experiments. Error bars represent SD. The P value is based on unpaired two-sided t test with 95% confidence. Source data are provided as a Source Data file.

Movie 3). The presence of micronuclei suggests that morphants and biallelic knockout embryos contain aneuploid cells.

We next checked whether the chromosome segregation errors observed in morphants were rescued by the co-injection of mRNA encoding zfTTLL11. Indeed, the proportion of anaphase cells with lagging chromosomes was reduced in these embryos, from 49.1% (in morphants) to 13% (n = 39 cells) (MO-1 vs MO-1+WT; P value <0.001; chi-squared test). In stark contrast, co-injection of mRNA encoding the catalytically dead zfTTLL11(E466G) (Supplementary Fig. 4c) did not rescue the chromosome segregation defects: 45% of anaphase cells showed at least one lagging chromosome (n = 31 cells) (MO-1 vs MO-1 + E446G; P value = 0.54; chi-squared test) (Fig. 3b).

Altogether, our data showed that TTLL11 is required for chromosome segregation fidelity in early zebrafish embryos. Since we found that its catalytic activity is essential, we conclude that polyglutamylation of the spindle MTs by TTLL11 is essential for error-free embryonic mitosis.

## TTLL11 is consistently downregulated in human tumors

The gain or loss of one or more chromosomes in daughter cells arises from chromosome mis-segregation events, generating aneuploidy, a hallmark of cancer cells. Approximately 86% of solid tumors in humans are aneuploid[22], and many cancer cells mis-segregate chromosomes at very high rates characteristic of chromosomal instability (CIN). Since our results showed that TTLL11 downregulation generates aneuploidy in the daughter cells (as detected by chromosome segregation defects, including lagging chromosomes and micronuclei), we decided to explore whether TTLL11 activity is compromised in cancer cells. We first explored whether the TTLL11 gene had mutations in human cancers that could potentially account for changes in its activity. In fact, we found that the rate of mutations in the coding and the untranslated region of the TTLL11 gene in cancer cells is significantly lower than expected (Supplementary Fig. 6a–e). We then looked for changes in TTLL11 expression levels in human cancers relative to the corresponding untransformed tissues. Strikingly, we found that TTLL11 is

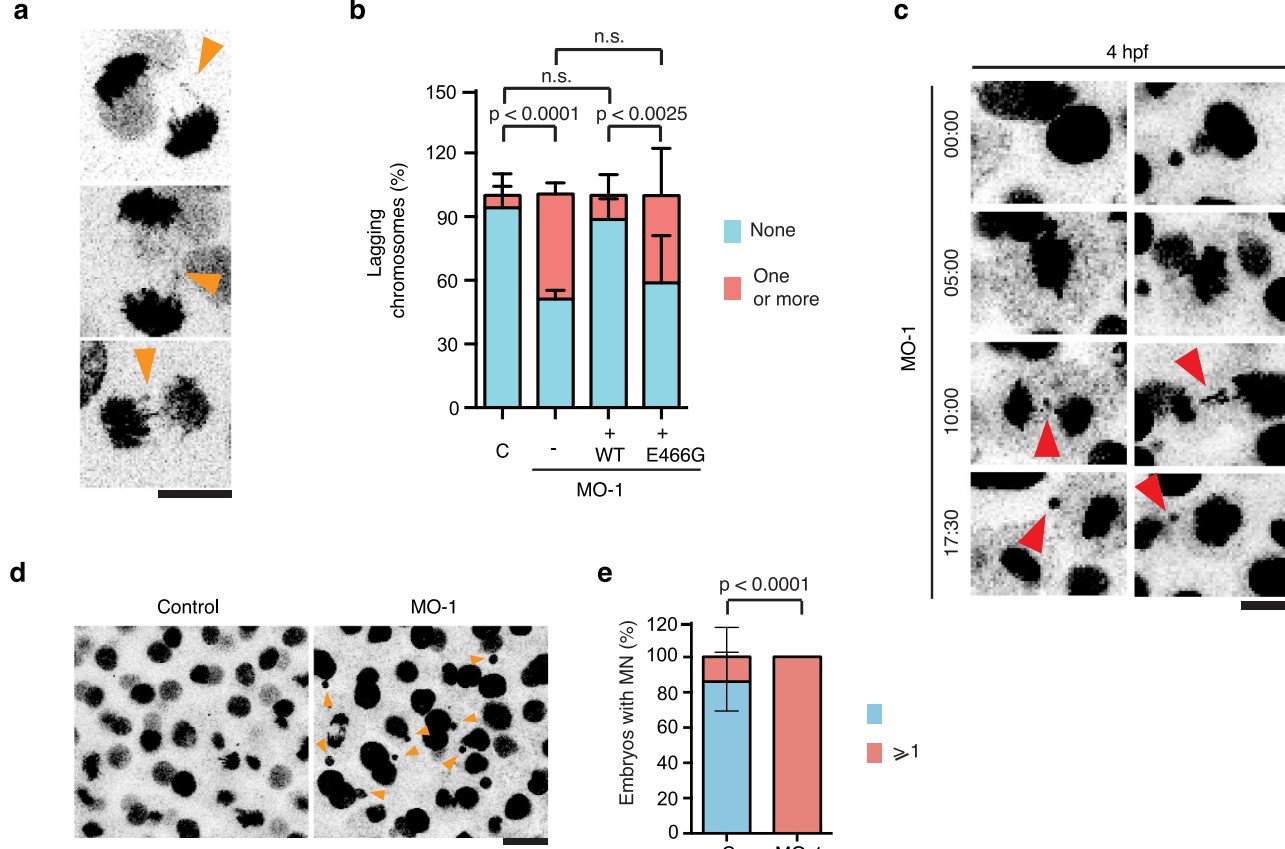

**Fig. 3 | Spindle MTs polyglutamylation by TTLL11 is required for chromosome segregation fidelity in zebrafish embryos. a** Selected images from movies of developing zfTTLL11-MO embryos showing several chromosome segregation defects in anaphase cells. Scale bar, 10 μm. **b** Quantification of anaphase cells with lagging chromosomes in control and morphant embryos expressing zfTTLL11 (+WT) or the catalytically dead zfTTLL11 (+E466G). n (control) = 41 embryos, n (morphant) = 46 embryos, n (+WT) = 39 embryos and n (+E466G) = 31 embryos. The total n is calculated over N = 3 independent experiments. P values based on a two-tailed $\chi^2$ test with a 95% confidence interval. Error bars represent SD. **c** Still images from movies of control and zfTTLL11-MO embryos expressing H2B-mCherry and showing lagging chromosomes forming micronuclei (red arrowheads) in the morphant cells. Time is mm:ss. Scale bar, 20 μm. **d** Still images from movies of control and zfTTLL11-MO embryos expressing H2B-mCherry and showing micronuclei (orange arrowheads) in the morphant cells. Scale bar, 20 μm. **e** Quantification from movies of developing zfTTLL11-MO embryos showing at least one event of a lagging chromosome generating a micronucleus at the end of cell division. n (control) = 20 embryos and n (morphant) = 22 embryos coming from N = 3 experiments. Error bars represent the SD. P value based on a two-tailed $\chi^2$ test with a 95% confidence interval. Source data are provided as a Source Data file.

significantly downregulated in all the tumors that have been characterized in the TCGA database as compared to the corresponding normal tissues (P value <0.0001; Wilcoxon rank-sum tests) (Fig. 4a and Supplementary Table 1). We also found that the consistent downregulation of TTLL11 in tumors is unique for this enzyme within the TTLL family; e.g., it was not observed for any of the other TTLL glutamylases (Supplementary Fig. 6f; P value = $6.3 \times 10^{-7}$; binomial test). Further, we found a clear negative correlation between TTLL11 expression levels and aneuploidy in cancer[23] (see "Methods") (Fig. 4b). The only other TTLL enzyme for which we found a clear trend was TTLL13 which is significantly overexpressed in the majority of tumors (Supplementary Fig. 6f). However, as TTLL13 basal expression levels are either low or undetectable in normal tissues, its overexpression in human cancer results in TTLL13 expression levels that are well below those of downregulated TTLL11 (Supplementary Fig 6g). Moreover, there was no correlation between the expression level changes of TTLL11 and TTLL13 in tumors suggesting they do not complement each other (Supplementary Fig 6g). Altogether, these data suggest that the downregulation of TTLL11 in human cancer is highly specific and therefore relevant functionally.

TTLL11 downregulation in human cancer should translate into reduced levels of spindle MT polyglutamylation in cancer cells. To test this hypothesis, we quantified the PolyE signal normalized by the total tubulin signal in spindles assembled in five human cancer cell lines: two colon cancer cell lines: HT-29 and HCT-116; two breast cancer cell lines: MDA-MB-231 and MDA-MB-468; and the osteosarcoma cell line, U2OS (Fig. 5a, b). We used as standard the normalized PolyE signal in spindles assembled in control and siTTLL11 hTERT-RPE1 cells (Fig. 5c). We found that the normalized PolyE levels were systematically lower in spindles from any of the cancer cell lines compared to those from control RPE1 cells. These reduced levels were similar to those of spindles from siTTLL11 RPE1 cells (Fig. 5c).

Altogether, these data show that the systematic downregulation of TTLL11 in human cancers correlates with the reduction of polyglutamylation of the spindle MTs that may play a role in favouring chromosome mis-segregation, aneuploidy, and CIN in tumor cells.

## TTLL11 downregulation is part of a cancer signature unique amongst this enzyme family

Multiple molecular mechanisms may underpin the consistent downregulation of TTLL11 across tumors. First, we hypothesized that tumor-specific promoter methylation of TTLL11 could inhibit its transcription. However, differences in promoter methylation levels between solid healthy tissues and primary tumor samples could not explain in all cases the transcriptional trend observed above (Supplementary Fig. 7).

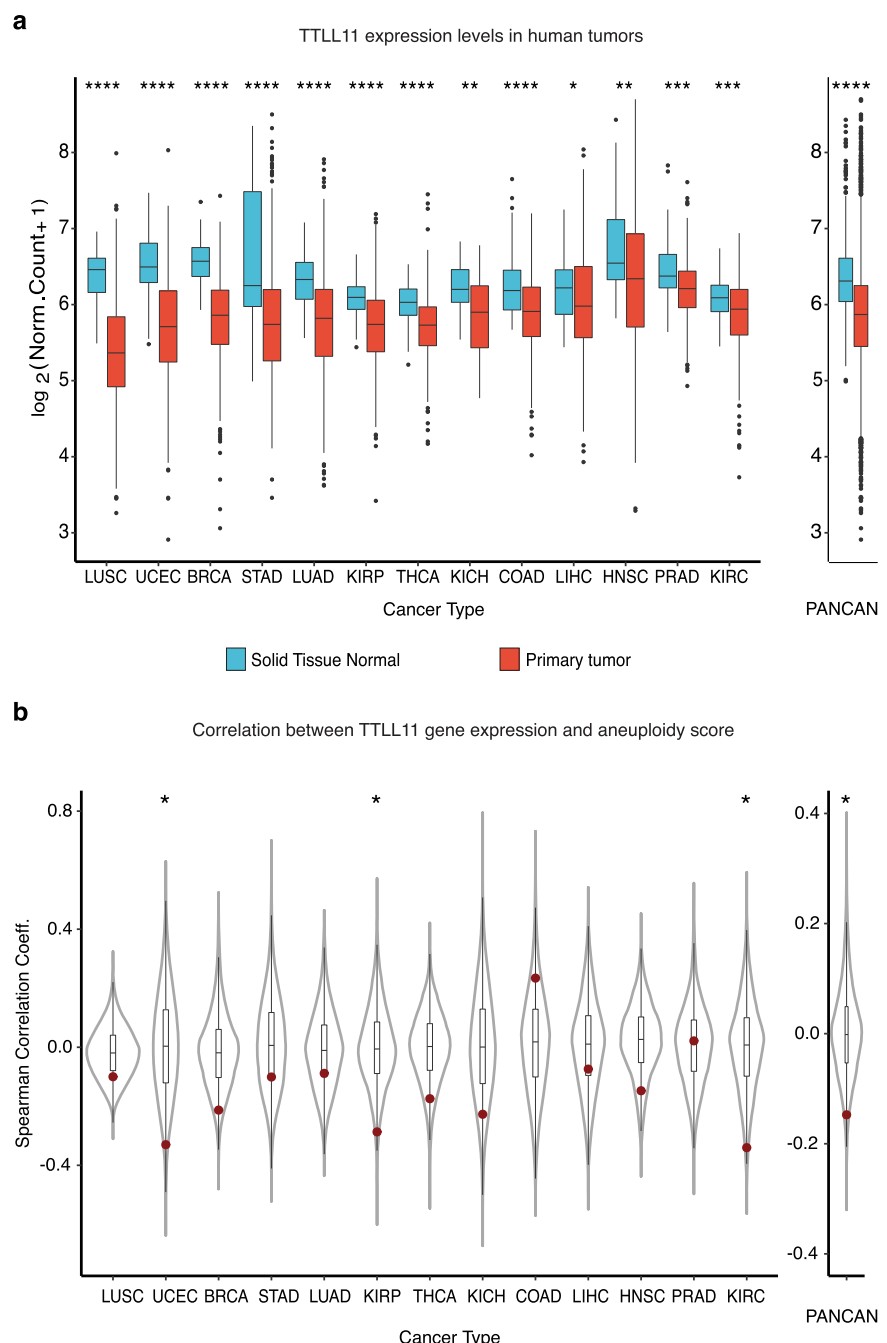

**Fig. 4 | TTLL11 expression is downregulated in tumors and correlates negatively with aneuploidy scores. a** Normalized expression of TTLL11 in healthy tissues and the corresponding primary tumor samples across 13 different cancer types, separately and combined (PANCAN). Nominal *P* values: ****$P \leq 0.0001$; ***$P \leq 0.001$; **$P \leq 0.01$; *$P \leq 0.05$; based on unmatched two-sided Wilcoxon rank-sum tests. The number of samples per cancer type and sample type (see Supplementary Table 3). In each box plot, the median value is indicated as a horizontal line and the lower and upper bounds of the box correspond to the first- and third quartiles, respectively. The upper- and lower whiskers range from the corresponding box hinges to the largest value no further than 1.5 times the interquartile range from the hinge. All outlying data points beyond the whiskers are plotted individually. **b** Spearman correlation coefficients of normalized gene expression and sample aneuploidy scores for every gene across 13 different cancer types. PANCAN represents the median correlation coefficient for every gene across cancers. TTLL11 is highlighted in red. Nominal *P* values: *$P \leq 0.05$, based on one-sided, one-sample *Z* tests. The number of genes detected per cancer type (see Supplementary Table 4). In each box plot, the median value is indicated as a horizontal line and the lower and upper bounds of the box correspond to the first and third quartiles, respectively. The upper- and lower whiskers range from the corresponding box hinges to the largest value no further than 1.5 times the interquartile range from the hinge. All outlying data points beyond the whiskers are plotted individually.

Alternatively, we hypothesized that the mRNA levels of TTLL11 may be regulated as part of a tumor-specific transcription program. To study this, we correlated the mRNA levels of each TTLL glutamylase against the mRNA levels of the rest of the genes across tumor and healthy samples, separately (see "Methods"). The resulting genome-wide co-expression profiles spotlighted TTLL11 as the TTLL glutamylase with the strongest tumor-specific co-expression signature that deviates from the expression pattern found in the corresponding normal tissues (Supplementary Fig 8a). In line with this, we found that the genes with altered co-expression with TTLL11 take part in biological processes directly related to mitosis (Supplementary Fig. 8b). We then asked which transcription factors could control this tumor-specific

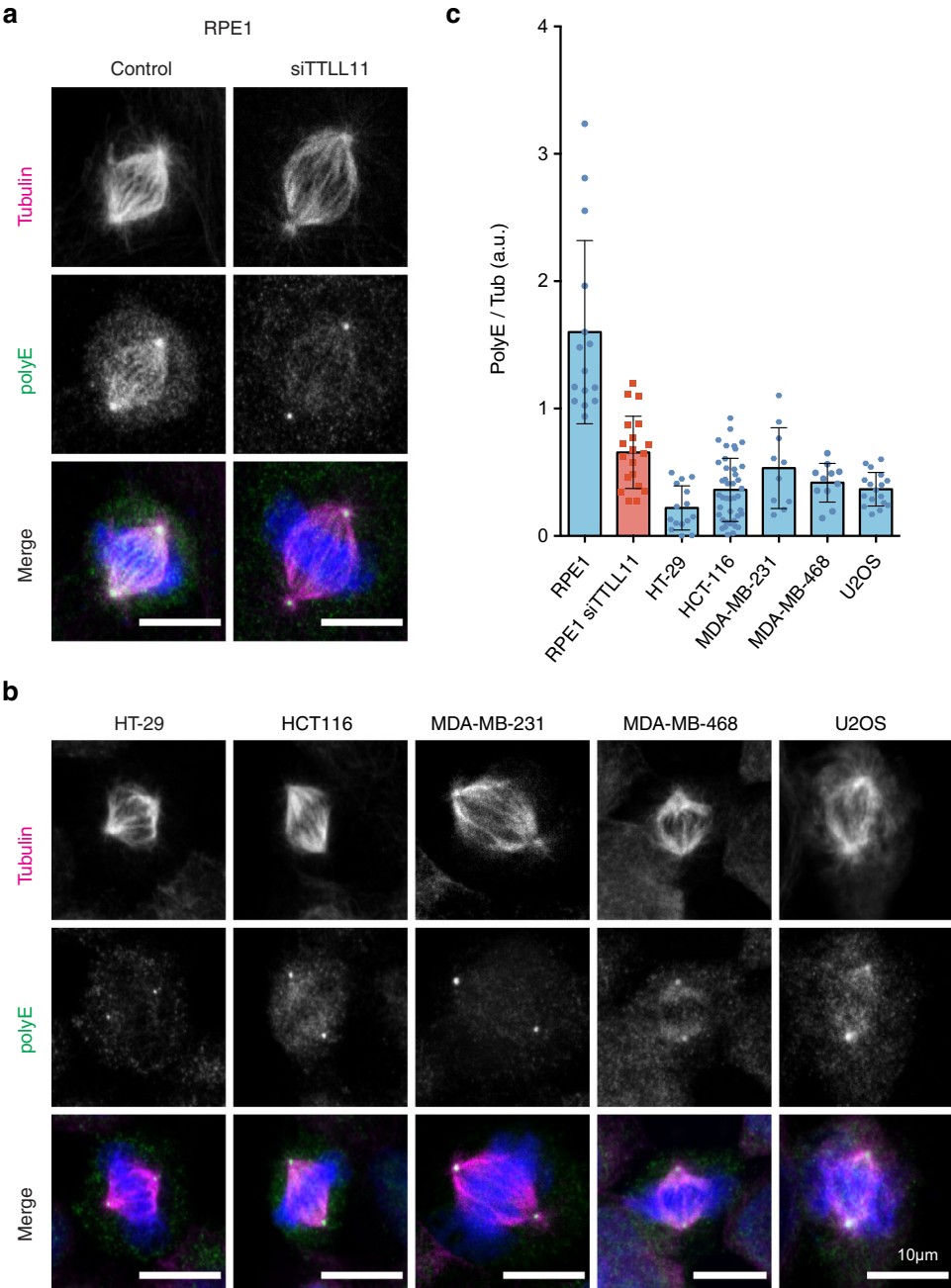

**Fig. 5 | Spindle MT polyglutamylation levels are reduced in cancer cells.**
**a** Immunofluorescence images of metaphase spindles in control and siTTLL11 hTERT-RPE1 untransformed cells, showing the PolyE signal (green), tubulin (red) and DNA (blue). Scale bars, 10 μm. **b** Immunofluorescence images of metaphase spindles in a panel of cancer cell lines as indicated. The PolyE signal (green), tubulin (red), and DNA (blue) are shown. Scale bars, 10 μm. **c** Quantification of the polyE signal normalized to the total tubulin signal in spindles from hTERT-RPE1 and cancer cells shown in (**b**). $n$ (RPE1) = 15 cells, $n$ (RPE1 siTTLL11) = 20 cells, $n$ (HT-29) = 15 cells, $n$ (HCT-116) = 41 cells, $n$ (MDA-MD-231) = 10 cells, $n$ (MDA-MD-468) = 11 cells and $n$ (U2O2) = 17 cells. Data are presented as mean values +/− SD. Source data are provided as a Source Data file.

transcription program. To do so we looked into the ChIP-seq-based ChEA dataset[24] (Fig. 6a) (see "Methods"). We found that genes with altered co-expression with TTLL11 are enriched in targets of the transcription factors NANOG (total targets = 6491), KDM5B (total targets = 3555), and ASH2L (total targets = 3127). Hence, the activity of these transcription factors may directly or indirectly control the mRNA levels of TTLL11. In a direct manner, only NANOG, a known cancer inducer[25], directly binds close to TTLL11 transcription start site. However, its weak co-expression with TTLL11 in tumors (Spearman = −0.02) suggests that the regulation of TTLL11 mRNA levels may rather be indirectly mediated by one or more of the genes targeted by the three

transcription factors (Fig. 6b). Among the 119 genes targeted by the three transcription factors, CCNE1 (also known as CyclinE), an oncogene involved in the control of mitosis frequently overexpressed in cancer (Fig. 6c)[26] was one of the top ten genes coexpressing with TTLL11. Recently, Guerrero and coworkers[27] quantified the transcriptomic changes upon overexpression of the oncogenes CCNE1, CDC25A, and MYC in non-transformed hTERT-RPE1 cells and in three different cancer cell lines (BT549, HCC1806, MDAMB231). Exploring their data, we found that TTLL11 mRNA levels decreased upon overexpression of CCNE1 and also of CDC25A, another oncogene targeted by two of the three transcription factors we identified above (NANOG

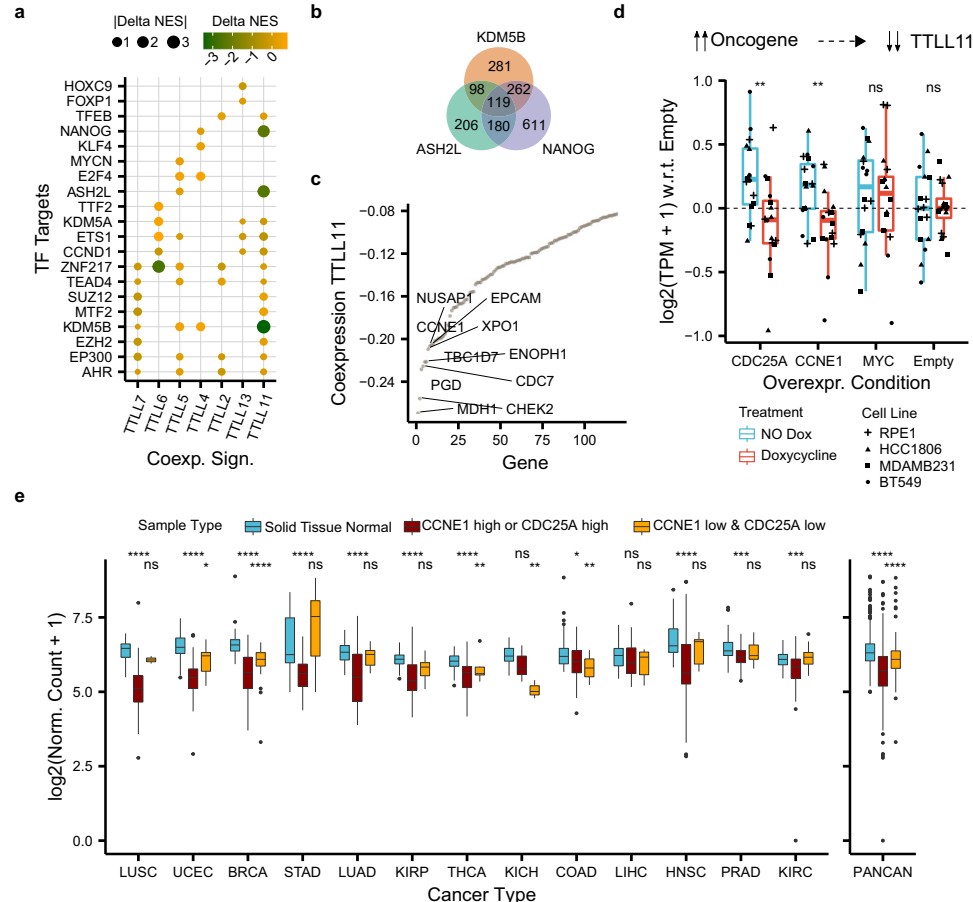

**Fig. 6 | Co-expression-based analysis pinpoints the overexpression of oncogenes CCNE1 or CDC25A as a cause for the consistent downregulation of TTLL11 across tumors. a** Differences in normalized enrichment scores (NES) (Delta NES) between the tumor and healthy co-expression signatures (*x* axis) of each TTLL glutamylase enzyme correlated against the rest of the genes. NES were obtained through gene-set enrichment analysis (GSEA) and show whether a co-expression profile is significantly enriched (FDR < 0.05) in genes targeted by a transcription factor (*y* axis) from the ChIP-seq-based ChEA dataset[24]. **b** Overlap between enriched transcription factor targets coexpressing with TTLL11 in tumors. **c** Co-expression (*y* axis) between TTLL11 and enriched 119 transcription factor targets of KDM5B, ASH2L, and NANOG in tumors (*x* axis). **d** Normalized gene expression of TTLL11 with respect to the control ("Empty") condition (*y* axis) upon overexpression with doxycycline ("Treatment" color) of different oncogenes (x-axis) across four different cell models ("Cell Line" dot shape). Each condition and treatment contain *N* = 16 biologically independent experiments (four for each cancer cell line). In each box plot, the median value is indicated as a horizontal line and the lower and upper bounds of the box correspond to the first and third

quartiles, respectively. The upper and lower whiskers range from the corresponding box hinges to the largest value no further than 1.5 times the interquartile range from the hinge. All outlying data points beyond the whiskers are plotted individually. **e** Normalized expression of TTLL11 (*y* axis) in healthy solid tissues and the corresponding primary tumor samples stratified by the expression of CCNE1 and CDC25A ("Sample Type" color) across 13 different cancer types (*x* axis), separately and combined (PANCAN). Statistical tests compare healthy solid tissues against either samples with high expression of CCNE1 or CDC25A ("CCNE1_high or CDC25_high") or samples with low expression of both upstream regulators of TTLL11 ("CCNE1_low & CDC25A_low"). Number of samples per cancer type and sample type (see Supplementary Table 5) In each box plot, the median value is indicated as a horizontal line and the lower- and upper bounds of the box correspond to the first- and third quartiles, respectively. The upper and lower whiskers range from the corresponding box hinges to the largest value no further than 1.5 times the interquartile range from the hinge. All outlying data points beyond the whiskers are plotted individually. In all the figure, nominal *P* values: ****≤0.0001, ***≤0.001, **≤0.01, *≤0.05, based on unmatched two-sided Wilcox rank-sum tests.

and KDM5B) and with altered co-expression with TTLL11 (Fig. 6d). Conversely, overexpressing MYC, an oncogene targeted by NANOG without altered co-expression with TTLL11, did not lead to consistent transcriptomic changes in TTLL11 (Fig. 6d). We then explored whether CCNE1 and CDC25A levels may also correlate with the mRNA levels of TTLL11 in patients. These two potential TTLL11 upstream regulators are actually often co-expressed in most types of cancers (Supplementary Fig. 8c). Stratifying patients according to the combined expression of CCNE1 and CDC25A revealed the expected pattern across most types of cancer: TTLL11 is downregulated in tumors with high expression of at least one of the two potential TTLL11 upstream regulators (Fig. 6e). Altogether, our analysis suggests that the consistent downregulation of TTLL11 across tumors may be explained by the overexpression of CCNE1 or CDC25A as part of a tumor-specific transcriptional program.

**Polyglutamylation defines the dynamic properties of spindle MTs**
We next addressed the mechanism underlying the critical role of spindle MT polyglutamylation in chromosome segregation fidelity. Chromosome segregation errors can occur through different mechanisms, including multipolar spindle formation, a weakened spindle assembly checkpoint, defects in kinetochore–MT attachments, error correction, and a partial MT stabilization[4,22,28]. Of note, we did not observe multipolar spindles in TTLL11-silenced cells or in morphants zebrafish embryos, ruling out that the observed chromosome segregation defects derived from defects in spindle geometry. We went on to determine whether a weakened SAC activity could account for the chromosome segregation errors in siTTLL11 cells. Like control cells, TTLL11-silenced cells incubated with an Eg5 inhibitor (*S*-trityl-l-cysteine, STLC) arrested in mitosis with monopolar spindles that accumulate a high number of erroneous syntelic attachments

occurring when sister kinetochores are linked by MTs to the same pole[29,30]. To assess the integrity of the kinetochore–MT attachment error correction mechanism, we measured the time required for cells to enter anaphase upon STLC washout. Live-cell imaging showed that siTTLL11 cells entered anaphase with kinetics very similar to that of the control cells, suggesting that the mechanism for error correction was fully functional in the absence of TTLL11 (191.6 ± 51.70 min in control cells vs 195.7 ± 49.84 min in silenced cells; $P$ value = 0.66; unpaired $t$ test) (Supplementary Movie 5).

We then focused on any potential changes in MT dynamics since a small increase of MT stability and/or assembly rate has been shown to compromise chromosome segregation fidelity[6,9,10,30]. Indeed, we noticed that spindles assembled in TTLL11-HeLa silenced cells appeared to be larger than those of control cells. Quantification showed that they were indeed longer (10.7 ± 0.8 μm versus 9.2 ± 1.0 μm for control cells; $P < 0.0001$) and wider (8.6 ± 0.9 μm versus 7.9 ± 0.6 μm for control cells; $P < 0.0001$) than controls (Fig. 7a). Consistently, spindles assembled in TTLL11-silenced hTERT-RPE1 cells and in morphant zebrafish embryos were also longer than controls (Fig. 7b, c). Altogether, this suggested that MTs may be partially stabilized in the absence of TTLL11. In agreement with this idea, a cold-stability assay showed that spindle MTs were more resistant to cold-induced MT depolymerization in TTLL11-silenced cells (Fig. 7d and Supplementary Table S3). Moreover, the interkinetochore distance in TTLL11-silenced cells was significantly reduced as compared to control cells (1.0 ± 0.3 μm versus 1.2 ± 0.3 μm in control; $P < 0.0001$) (Fig. 7e), further suggesting that MTs were partially stabilized[31]. Altogether, these data consistently pointed to an increased stability of the spindle MTs in the absence of TTLL11.

To gain additional support for this idea, we measured the microtubule poleward flux in TTLL11-silenced cells using a stable HeLa cell line expressing H2B–RFP/PA-GFP–tubulin. We found that the flux rate was significantly reduced in the absence of TTLL11 as compared to control cells (0.4 ± 0.1 μm/min versus 0.6 ± 0.1 μm/min in control; $P < 0.0002$) (Fig. 7f and Supplementary Movie 6). Altogether, our data consistently showed that MTs are less dynamic in the absence of TTLL11, revealing that MT polyglutamylation provides a mechanism to establish finely-tuned spindle MT dynamics and ensure faithful chromosome segregation.

## Discussion

Here, we have uncovered a previously unrecognized mechanism that ensures correct chromosome segregation fidelity that appears to be disrupted in human cancers. (Poly)glutamylation is one of the most prevalent tubulin PTMs[11]. Although polyglutamylation of the spindle MTs was reported several years ago[32,33], the functional relevance of this PTM in mitosis had not been addressed so far. Polyglutamylation is catalyzed by TTLL family members that can act as initiating or elongating enzymes and generate different levels and patterns of polyglutamylation on MT networks[11]. Here we found that within the TTLL glutamylase enzyme family, only TTLL11 and TTLL13 (both previously characterized as elongases[17]) localized to the mitotic spindle when expressed as GFP fusions. Consistently, we found here that TTLL11 drives the assembly of long glutamate chains (of three or more glutamates) specifically on the spindle MTs. We do not know yet the mechanism providing this high specificity for the spindle MTs. Overexpressed recombinant TTLL11 localizes to the cytoplasm and the nucleus both in interphase HeLa cells and zebrafish embryos. In the cytoplasm overexpressed TTLL11 is highly active on interphase MTs suggesting that it is an autonomous enzyme that does not require additional factors for activation. By contrast, the endogenous enzyme may be unable to polyglutamylate the interphase MTs because it is nuclear. The nuclear localization and specific function of TTLL11 on the spindle MTs suggest that it may be a novel SAF (Spindle Assembly Factor), a

group of nuclear MT-associated proteins with specific functions in spindle assembly that are regulated by RanGTP around the chromosomes[34,35].

Impairing MT polyglutamylation in mitotic cells by down-regulating TTLL11 does not preclude bipolar spindle assembly or chromosome segregation. However, it is not without consequences. Indeed, our results consistently show that it increases the rate of lagging chromosomes in anaphase, both in vitro and in vivo. Although previous studies showed that a majority of anaphase lagging chromosomes segregate to the correct daughter cell[36], this is not the case here. Indeed, we found that many cells in zebrafish embryos lacking TTLL11 or expressing a catalytically dead TTLL11 enzyme contained micronuclei, a hallmark of chromosome mis-segregation events. The resulting cell aneuploidies most certainly explain the increase in early embryonic death and the dramatic developmental defects of these embryos.

The mechanism at play involves a change in the dynamics of the spindle MTs. Indeed, the increased spindle size and the increase resistance of the spindle MTs to cold-induced depolymerization in TTLL11-silenced cells all indicate that the spindle MTs are partially stabilized in these cells. Although the mechanism driving the spindle flux is still not fully understood it involves a fine balance between microtubule sliding and dynamics[37,38], the reduced speed of the spindle flux in TTLL11-silenced cells is also compatible with a reduced dynamics of the spindle MTs. Previous studies showed that a slight stabilization of the spindle MTs leads to chromosome segregation errors[6,9,10,30]. In the absence of any other defect at the level of other MT PTMs such as tyrosination/detyrosination or acetylation, SAC activity, bipolar spindle assembly and geometry and kinetochore attachment error correction mechanism, the partial stabilization of the spindle MT that we observed in the absence of TTLL11 may therefore fully account for the chromosome mis-segregation events that we observed in HeLa and RPE1 cells and in zebrafish embryos. Our data therefore suggest that MT polyglutamylation provides normal cells with a mechanism that, beyond the SAC, protects cells from making errors during chromosome segregation.

MT (poly)glutamylation has been shown to regulate the recruitment and activity of a number of MT-interacting proteins, including MAPs, severing enzymes and molecular motors[11,39,33,40–46]. We did not detect significant changes in the localization of a number of mitotic MAPs in silenced cells. This may not be unexpected because bipolar spindles do assemble and segregate chromosomes in the absence of TTLL11 albeit segregation fidelity is compromised. Overall, our data suggest that the polyglutamylation of the spindle MTs can act as a central regulator defining the binding affinities and/or activities of a number of interacting proteins that altogether can account for the fine regulation of MT dynamics in the spindle.

This mechanism for ensuring chromosome segregation fidelity in normal cells is compromised in human cancer, in which TTLL11 is systematically downregulated. Consistently with this downregulation, we found that the level of spindle MT polyglutamylation is reduced in a panel of human cancer cell lines. Interestingly, our data showing that the downregulation of TTLL11 result in a partial stabilization of the spindle MTs is consistent with the observation that cancer cells with CIN have hyperstable kinetochore–MT attachments as compared to stable diploid cells[9,22,29,30].

The relevance of TTLL11 downregulation in cancer is underscored by our findings revealing that it is the only TTLL glutamylase expressed in all tissues with a tumor-specific co-expression signature. Although we could not identify a specific transcription factor that may be directly responsible for the downregulation of TTLL11 in cancer we found that there is a causal link between the increased expression of two oncogenes, CCNE1 and cdc25a, and the downregulation of TTLL11. Altogether, these data corroborate the significance of TTLL11 downregulation in cancer.

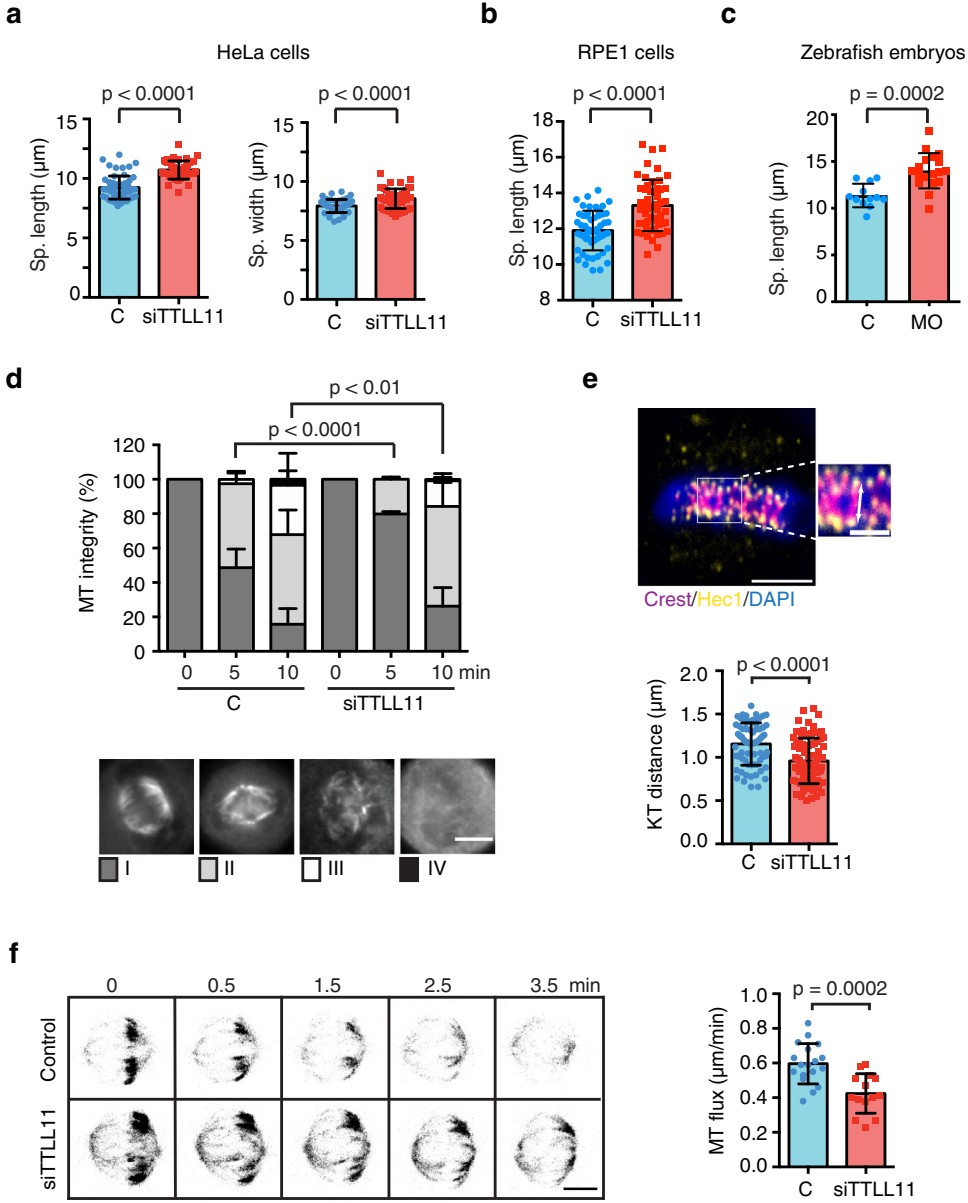

**Fig. 7 | TTLL11 silencing results in the partial stabilization of the spindle MTs.**
**a** Quantification of the spindle length and width in control or siTTLL11 HeLa cells.
Length: *n* (control) = 55 cells and *n* (siTTLL11) = 45 cells. Width: *n* (control) = 47 cells
and *n* (siTTLL11) = 47 cells. Graphs show one representative experiment of *N* = 4
independent experiments. Data are presented as mean values +/− SD.
**b** Quantification of the spindle length in control and siTTLL11 hTERT-RPE1 cells. *n*
(control) = 53 cells and *n* (siTTLL11) = 53 cells. Graphs show one representative
experiment of *N* = 3 independent experiments. Data are presented as mean
values +/− SD. **c** Quantification of the spindle length in control and MO-1 zebrafish
embryo dissected cells. *n* (control) = 28 cells and *n* (MO) = 26 cells. Graphs show
one representative experiment of *N* = 2 independent experiments. Data are pre-
sented as mean values + /− SD. **d** Quantification of cold-induced K-fibers depoly-
merization over time in control and siTTLL11 cells. Cells were classified into four
categories as shown. *n* (control[0]) = 100 cells, *n* (control[5]) = 87 cells, *n* (con-
trol[10]) = 87 cells, *n* (siTTLL11[0]) = 100 cells, *n* (siTTLL11[5]) = 105 cells and *n*

(siTTLL11[10]) = 115 cells. *P* values based on $\chi^2$ test with a 95% confidence interval.
Scale bar, 15 μm. **e** Immunofluorescence image of metaphase plate aligned chro-
mosomes in HeLa cells, showing chromosomes (blue), CREST (magenta, kine-
tochore), and Hec1 (yellow). In the magnification, the white arrow shows the
measured interkinetochore distance. Scale bar, 5 μm; for zoom, 2 μm. Quantifica-
tion of interkinetochore distance. For si-control, *n* = 72 kinetochore pairs, from 16
cells; for siTTLL11, *n* = 82 kinetochore pairs, from 18 cells. Scale bar, 5 μm; in zoom,
2 μm. **f** Confocal images of tubulin photoactivated (dark gray) close to the meta-
phase spindle equator (0 min) over time (min) in control or siTTLL11 cells. The
velocity of the poleward flux in control cells (*n* = 18) and siTTLL11 cells (*n* = 14). Scale
bar, 15 μm. All graphs correspond to three independent experiments. Data are
presented as mean values +/− SD. Unless otherwise indicated *P* values are based on
an unpaired two-sided *t* test with 95% confidence. Source data are provided as a
Source Data file.

In summary, by identifying TTLL11 as the polyglutamylase that
modifies spindle MTs, we discovered an essential role for this PTM
in chromosome segregation fidelity. Altogether, our data showed
that TTLL11-dependent polyglutamylation of MTs provides a
central mechanism beyond the SAC that could explain why cells
rarely mis-segregate chromosomes in normal tissues[16]. The

systematic downregulation of TTLL11 in cancer cells underscore
the functional relevance of MT polyglutamylation in the spindle.
Further work will determine whether cancer cells exploit this
mechanism to promote chromosome segregation errors, aneu-
ploidy and CIN, and whether it can provide novel targeted
approaches for cancer therapies.

## Methods

### Cell lines and plasmids

All the human cell lines (HeLa, hTERT-RPE1, HT-29, MDA-MB-231, MDA-MB-468, HCT-116, and HEK-293T, U2O2) were grown at 37 °C in a 5% $CO_2$ humid atmosphere in Dulbecco's Modified Eagle Medium (DMEM) supplemented with 4.5 g/L glucose and 4.5 g/L glutamine, 10% fetal bovine serum, 100 units/ml penicillin and 100 µg/ml streptomycin. Stable HeLa cell lines expressing H2B–mRFP/α-tubulin–GFP and H2B–mRFP/PA-α-tubulin–GFP (a kind gift from P. Meraldi ETH, Zurich) were grown in the presence of 400 µg /ml G418 and 20 µg/ml puromycin.

### Plasmid and RNAi transfection

To express fluorescently-labeled proteins, cells were transfected for 48 h using Lipofectamine 2000 (Invitrogen) with 1 µg of human TTLLs tagged with GFP at either end. TTLLs' cDNA of TTLL1, 4, 5, 6, 7, 9, 11, and 13 were obtained from human adult brain tissue and cloned into a peYFP-C1 vector for mammalian expression[1]. All cDNAs were then subcloned into the peGFP-N1 and peGFP-C1 vectors. TTLL2 and TTLL12 cDNAs were obtained from the human ORFeome collection of the CRG Protein Facility Unit. TTLL2 and TTLL12 cDNAs were also subcloned into peGFP-N1 and peGFP-C1 vectors.

To perform RNA interference (RNAi), HeLa and hTERT-RPE1 cells were transfected at a 50–60% confluence using 500 pM RNAiMAX (Invitrogen) with 100 nM siRNAs for 48 h. An siRNA smart pool (Dharmacon, L-016360-01-0020).against hTTLL11 was used with the following sequences: TTLL11#1, 5′-UGACGGAGAUGGUGCGUAA-3′; TTLL11#2, 5′-GAGUUUCAUUUCACGACAA −3′; TLL11#3 5′-UCAAAUG-GUGAAAGACGAU −3′; TTLL11#4, 5′-GGAUUCUGCCUGACGAGUU-3′.

### SDS-PAGE and western blot analysis

For western blot analysis, the following primary antibodies were used: rabbit anti-polyE antibody (polyE, made in-house, 1.1 µg/µl), diluted 1:1000; mouse anti-α-tubulin (DM1A, Sigma T9026), diluted 1:1000; and rabbit anti-GFP (GFP, made in-house, 0.6 µg/µl), diluted 1:500. Western blots were developed with Odyssey CLx imaging system (LI-COR Bioscience). Uncropped scans are provided in the Source Data file.

### Immunofluorescence microscopy

HeLa cells were grown on glass coverslip and fixed in ice-cold methanol for 3 min at −20 C°. The following primary antibody were used: rabbit anti-polyE antibody (polyE, made in-house, 1.1 µg/µl) diluted 1:1000, mouse anti-α-tubulin (DM1A, Sigma T9026), diluted 1:1000; rabbit anti-GFP (GFP, made in-house, 0.6 µg/µl), diluted 1:1000; rabbit anti-β-tubulin (Abcam ab6046), diluted 1:500; rabbit anti-detyrosinated tubulin (ΔTyr, Sigma-Aldrich AB3201) diluted 1:500; rat anti-tyrosinated tubulin (YL1/2, Sigma-Aldrich MAB1864) diluted 1:250; mouse anti-acetylated tubulin (Ac, Sigma-Aldrich T7451) diluted 1:500; mouse anti-glutamylated tubulin (GT335; Enzo 804-885), diluted 1:1000; human anti-centromere proteins (CREST, Antibodies Incorporated 15−235), diluted 1:100; and anti-Hec1 (Hec1, Genentek GTX70268), diluted 1:100. DNA was counterstained with DAPI (1 µg/ml; Sigma-Aldrich), diluted 1:1000. Antibodies were diluted in the following buffer: 1× PBS, 0.1% Triton-X-100 (v/v), 0.5% BSA (w/v). Images were acquired with a Leica DMI 6000b microscope mounted with a DF2 90000GT camera or with a Leica TCS SPE confocal microscope using the LAS X 4.13 software, processed with ImageJ 2.3.0 and the figures assembled in Adobe Illustrator CS5.1.

### Quantification of immunofluorescence samples

The length of HeLa cell metaphase spindles was obtained by manually tracing a line from pole-to-pole using ImageJ. ImageJ scales were checked for correct pixel/µm conversion. To quantify the level of tubulin (poly)glutamylation in interphase and mitotic cells, the signal intensity for the selected antibody was normalized with the DM1A tubulin signal. To limit signal fluctuation, the following formula was used:

$$Normalized\ Intensity_x = \frac{Raw\ Intensity_x - (Area{*}\overline{Noise}_x)}{Raw\ Intensity_{tub} - (Area{*}\overline{Noise}_{tub})} \quad (1)$$

where $x$ is the MT PTM of interest, Area is the circular ROI drawn around the metaphase spindle and must be identical for both signals, and $\overline{Noise}$ is the mean of the average intensity signal of three random areas within the cell around the spindle. The interkinetochore distance was obtained by manually tracing a straight line between sister chromatid centromeres detected with an anti-Hec1 antibody (Hec1, Genentek GTX70268) connected with CREST staining (CREST, Antibodies Incorporated 15−235) in metaphase spindles. Measurements were only validated for a given spindle if at least five different values could be obtained. The lagging chromosome frequency was assessed in fixed HeLa cells stained with DAPI (1 µg/ml; Sigma-Aldrich) diluted 1:1000 to detect DNA and centromeres with an anti-CREST antibody (CREST, Antibodies Incorporated 15−235) diluted 1:100; it was calculated by monitoring the presence of lagging chromosomes (e.g., showing both DNA and centromere positive signals) in-between the two main masses of separating chromosomes for at least 121 anaphase cells. Data were then analyzed with Prism6 (Graphpad).

### K-fiber cold stable assay

Hela cells were cultured on 18-mm round coverslips in DMEM. Cells were washed 3× for 5 min in 1× PBS. Medium was replaced by cold L15 medium at 4 °C, and cells were placed on ice. Coverslips were retrieved at given timepoints, and cells were fixed in ice-cold methanol at −20 °C for 3 min. Slides were stained with anti-α-tubulin (DM1A, Sigma T9026) diluted 1:1000, and for DNA (DAPI 1 µg/ml; Sigma-Aldrich) diluted 1:1000. The quantification of K-fiber stability upon cell incubation on ice over time was obtained by scoring the status of the k-fiber MTs among four arbitrarily defined classes.

### Live-cell imaging

HeLa cells stably expressing H2B–mRFP/α-tubulin–GFP were cultured in a 35/10 mm glass bottom, four-compartment dish (Grainer Bio-one, 267061) and imaged using a ×60 oil-immersion 1.4 NA objective on Andor Dragon Fly Spinning Disk confocal microscope. For imaging, media was replaced with fresh media without phenol red. Several random fields were selected for imaging to increase the possibility of visualizing mitotic events. Each field was imaged every 2 to 3 min over 6 h by taking a 15-µm Z-volume divided in 5 to 7 intervals (depending on the experiment). Movies were then processed using ImageJ (ref. NIH Image to ImageJ: 25 years of image analysis).

The STLC release experiment was performed by adding 10 µM STLC to growing HeLa cells for 2 h. STLC was removed by 4× washes in warm, 1× PBS and one wash with DMEM. Cells were placed under the microscope Andor Dragon Fly Spinning Disk microscope and imaged using a ×60 oil-immersion 1.4 NA objective. Each field was imaged every 2 to 3 min over 6 h by taking a 15-µm Z-volume divided in 5 to 7 intervals (depending on the experiment). The time required for cells to enter into anaphase was calculated starting from the time of the first wash. All displayed images represent maximum intensity projection of z-stacks. The Andor Dragon Fly system was equipped with an iXON-EMCCD Du-897 camera, and Andor iq 3 imaging software was used for images acquisition.

### RT-qPCR

For RT-qPCR in HeLa cell, mRNA was isolated with TRIzol Reagent (Invitrogen) from a normal, asynchronous Hela cell population. Total mRNA was quantified with a NanoDrop spectrophotometer

and retro-transcribed into cDNA with the Superscript III (Invitrogen, 12574-026). cDNA was used for quantitative PCR with reverse transcription (RT-qPCR) analysis using SYBR green (Thermo-Fischer). Oligonucleotide sequences are indicated in Supplementary Table 2.

## Tubulin poleward flux measurement

HeLa cells stably expressing H2B–RFP/PA-α-tubulin–GFP were cultured in a 35/10 mm glass bottom, four-compartment dish (Grainer Bio-one). For imaging, cells were kept at 30 °C using an Okolab stage top chamber (UNO-T-H-CO₂) and imaged using a 63× oil-immersion 1.4 NA objective lens on a Leica TCS SP5 confocal microscope. Images were acquired using the LAS X 4.13 software. Bipolar spindles were identified by looking at the H2B−mRFP signal. PA-GFP-α-tubulin was activated in thin stripes on the side of the metaphase plate with a 405 nm laser (100%) (1 to 2 μm wide to cover the width of the metaphase plate). GFP fluorescence was captured every 8–10 s for 270 s. The poleward MT flux rate was calculated by generating a kymograph of the fluorescent speckle[47] using ImageJ 2.3.0.

## Zebrafish husbandry, strains

Zebrafish (*Danio rerio*) were maintained as described in Westerfield, M. (2000). The zebrafish book. A guide for the laboratory use of zebrafish (Danio rerio). 4th ed., Univ. of Oregon Press. Wild-type embryos were obtained from the AB strain natural crosses and kept in an incubator at 28 °C until the sphere stage. All protocols used for the experiments were approved by the Institutional Animal Care and Use Ethic Committee (PRBB–IACUEC). Experiments were carried out in accordance with the principles of the 3Rs. The transgenic fish line Tg(*actb2*:h2amCherry) (Zfin: e103Tg) was used for in vivo imaging experiments.

## zfTTLL11 cloning, RT-PCR, and mutagenesis

Total RNA was isolated from 24 hpf zebrafish embryos using tripleX-tractor direct RNA (mirage biomedicals, GK23.0100) and reverse transcribed using a Xpert cDNA Synthesis kit (mirage biomedical GK80.0100).

zf_TTLL11 (ZDB-GENE-061013-747) cDNA was amplified using Phusion HF (Thermofisher F530S) and cloned into pCS2 vector (BamHI-EcoRI linearized) through Gibson cloning, using the following primers:

TTLL11-pCS2-Fw:

5′-CTACTTGTTCTTTTTGCAGGATCCATGAGCGATCACTACGAGAGAGT-3′

TTLL11-pCS2-Rv:

5′-GCTCGAGAGGCCTTGAATTCTCAGTTGTCTGTGTTGGCTTTAGCAG-3′

To visualize the protein, a Gibson system was used to subclone *ttll11* into a pCS2 vector containing a GFP at the N-terminal end (XhoI linearized), to obtain the GFP-zfTTLL11 fusion protein. The following primers were used:

TTLL11-GFP-GibpCS2-Fw:

5′-CAAGGAATTCAAGGCCTCTCGAATGAGCGATCACTACGAGAGAGT-3′

TTLL11-GFP-GibpCS2-Rv:

5′-ACTCACTATAGTTCTAGAGGCTCAGTTGTCTGTGTTGGCT-3′

Around 35 ng cDNA was used for each stage for RT-PCR (Tm = 57 °C; 33 cycles), using GoTaq DNA polymerase (Promega, M7805). Oligonucleotide sequences are indicated in Supplementary Table S4.

The pCS2-ttll11 construct was used to generate the mutated zfTTLL11-E466G version with QuickChange Site-Direct Mutagenesis kit (Agilent) with the following primers:

Fw: 5′-CTTGAAGCCTGTTTTTATTAGGAGTCAATGCCAATCCCAGC-3′

Rv: 5′-GCTGGGATTGGCATTGACTCCTAATAAAACAGGCTTCAAG-3′.

## Morpholinos, mRNA synthesis, and microinjection

Morpholino antisense oligonucleotides were designed and then purchased from Gene Tools, LLC. To inhibit *ttll11*, a blocking translation MO (CGGCTGATTTGTTATCTCATCTAGG) was used, with a standard control MO (CCTCTTACCTCAGTTACAATTTATA) used as a negative control. About 2.8 ng of morpholino was injected into each one-cell stage embryo.

All capped mRNAs were synthesized using mMessage mMachine SP6 (Ambion, AM1340M). For the rescue experiments, 200 pg of *ttll11* mRNA was injected together with the indicated MO; 200 pg E466G mRNA (*ttll11* mutated version) was injected together with the indicated MO as a negative control for the rescue experiments; GFP-Ttll11 mRNA was injected into each one-cell stage embryo to visualize protein localization.

A PV820 microinjector (WPI) combined with a M3301R micromanipulator was used to perform the microinjections.

## F0 biallelic knockouts in zebrafish

crRNA target sites guided to exon 2, 3 and 4 of the Ttll11 zebrafish gene (ENSDARG00000060374) were designed using the ChopChop web tool (https://chopchop.cbu.uib.no/)[48] and are listed in Supplementary Table 9. crRNAs, TracRNA (1072532), Alt-R® S.p. Cas9 Nuclease V3 (1081058) and the three Alt-R™ Crispr Negative Control crRnas (1072544, 1072545, 1072546) were all purchased from IDT (Integrated DNA Technologies). The assembly for all the components was made following the protocol published in ref. 19. Approximately 2 nl of the mix containing the three RNPs (gRNA assembled with Cas9) directed to Ttll11 and the negative control were injected into 1-cell stage embryos. Embryos were collected at 36 hpf for mRNA and genomic DNA extraction.

A total of 13 injected embryos and 3 WT embryos at 36 hpf were collected separately in a PCR tube containing 100 μl of 50 mM NaOH. After an incubation of 15 min at 95 °C, 10 μl of 1 M Tris-HCl was added. In all, 1 μl of this solution was directly used to perform the fluorescent PCR, as described in ref. 49. A common M13Fw universal primer (GTAAAACGACGGCCAGT) labeled with FAM was used for all the reactions. The PCR reaction was made using GoTaq polymerase (M7805, Promega) using the primers listed in the table. The PCR program was: 95 °C for 5 min; then 40 cycles of: 95 °C for 30 s, 60 °C for 30 s, 72 °C for 30 s; then 72 °C for 15 min. After running a 1% SB (Sodium Borate) agarose gel to verify the amplification (data not shown), the fluorescent PCR was analyzed using ABI 3130xl genetic analyzer (Life Technologies). ROX 500 (401734, Applied Biosystems) was used as a size standard.

## Expression analysis

A total of 20 embryos at 36 hpf were embedded in 700 μl of Trizol for each condition (injected and non-injected embryos). The total mRNA of every batch of embryos were extracted using TripleXtractor Direct RNA kit (GK23.0100, Grisp). cDNA was synthesized from 500 ng of the total mRNA extracted using SuperScript III (11752050 Life Technologies). In total, 50 ng of cDNA were used to perform a semiquantitative PCR using GoTaq polymerase (M7805, Promega). The eef1a1 zebrafish gene was used as a PCR control. The primers used were: eef1a1_qF, GACATCCGTCGTGGTAATG; eef1a1_qR, GATGATGACCTGAGCGTTG and zf_ttll11_qF, GTGGACATCAAGAAGGTCTG; zf_ttll11_qR, CAAAGCCCAGGATCTGAAA.

The PCR program was: 95 °C for 5 min; then 27 cycles of: 95 °C for 30 s, 60 °C for 30 s, 72 °C for 30 s; then 72 °C for 15 min. A 1.5 % of SB agarose gel was run 20 min at 120 V to analyze the semiquantitative PCR.

In total, 30 ng of cDNA were used to perform a qPCR using Pow-erUp™ SYBR™ Green Master Mix (A25742, Applied Biosystems). The eef1å1 zebrafish gene was used as a housekeeping gene. qPCR was run in a Geneamp 9700 (Applied Biosystems) thermocycler using the standard program. The results were analyzed using the ΔΔCt method.

## Zebrafish immunofluorescence

Zebrafish embryos at the sphere stage (2.5–4 hpf) were dechorionated and incubated overnight with room temperature (RT) shaking in the MT-fixative solution[50]. The MT-fixative was discarded, and embryos were put into methanol at −20 °C overnight for fixation. Embryos were then transferred to a clean tube and washed 3× for 5 min each with tergitol at room temperature for rehydration. Embryos were then moved to a 96-well plate and washed overnight with the anti-autofluorescence buffer (1× PBS, 100 mM NaBH$_4$) at room temperature with gentle shaking. Embryos were then washed 5× with 1× TBS followed by a blocking step with 1× TBS, 2% BSA for 30 min at room temperature with gentle shaking. The anti-polyE (1.1 μg/ml, made in-house) and anti-β-tubulin (clone E7; Hybridoma Bank at the University of Iowa) primary antibodies were added at a 1:200 dilution, and samples were incubated overnight at 4 °C with gentle shaking. After primary antibody incubation, samples were washed rapidly 5× with 1× TBS, secondary antibodies Alexa Flour (Invitrogen) were added at 8 μg/ml, and samples were incubated for 3 h at room temperature with gentle shaking. Embryos were then rinsed twice with 1× TBS and then incubated for 20 min with DAPI (1 μg/ml; Sigma-Aldrich) diluted 1:500, rinsed twice with 1× TBS and finally washed overnight at 4 °C in 1× TBS before mounting. Embryos were then transferred to a tube containing low-melting-point agarose at 42 °C (diluted in 1× TBS) and immediately placed into a Mattek dish with a 7-mm diameter glass bottom, and oriented with epithelial layer cells toward the glass slide, before the agarose solidified. Once the agarose had solidified, it was covered with 1× TBS to avoid evaporation.

## Fixed and live-imaging of zebrafish embryos

Fixed embryos were imaged in glass bottom Mattek dishes using a ×63 oil-immersion 1.4 NA objective on a TCS SP5 inverted Leica confocal microscope.

For zfTTLL11 localization experiments, zebrafish eggs were fertilized, collected, and microinjected with GFP-zfTTLL11 mRNA with a micromanipulator (M3301R, WPI World precision instruments). The quantity injected were relative to the ones specified in "Morpholinos, mRNA synthesis and microinjection" section, according to the experiment. Embryos at sphere stage were manually dechorionated and moved into a tube containing low-melting-point agarose at 42 °C diluted in 1× TBS and then immediately into a Mattek dish with 7-mm diameter glass bottom, with the epithelial layer of cells oriented toward the glass slide, before the agarose solidified. Embryos were imaged under an Andor Revolution HD Spinning Disk microscope with a 60×, 1.4 NA oil objective with 2-min time-lapse intervals, taking images every 1.5 μm in a 40–60 μm volume. An iXON- EMCCD Du-897 camera and Andor IQ Imaging software were used for image acquisition. For anaphase lagging chromosome experiments, zebrafish eggs from the Tg(bactin:H2AmCherry) line were fertilized, collected and microinjected with either scrambled or MO-1 (as in "Morpholinos, mRNA synthesis and microinjection"). Embryos were collected as above but laid on a 35/10-mm glass bottom, four-compartment dish (Grainer Bio-one). Embryos were imaged using a Leica TCS SP8 confocal microscope equipped with an Argon laser. An HC PLAN APO 63×1.4 NA objective was used. Images were acquired simultaneously using the LAS X 4.13 software every 45s-75s according to the experiments, imaging a 20 μm Z-volume every 2 μm. All live-imaging experiments were performed at room temperature.

## Zebrafish embryo dissection and fixation

Zebrafish embryos at the sphere stage (2.5–4 hpf) were dechorionated in glass dishes with 10 ml of E3 media. 30–50 embryos were transferred in a 1.5 ml Eppendorf tube. The excess of media was removed and 500 μL of cell culture media DMEM-F12 pH 7.4–7.5 (D3) was added. Embryo cell dissection was obtained by vigorously shaking the tube until no visible embryos are left at the bottom. The suspended cells and yolks are then centrifuged at 200×$g$ for 3 min, after centrifugation the supernatant is discarded and the cells are resuspended in other 500 μL of cell culture media D3 and centrifuged at 200×$g$ for 3 min. The supernatant is discarded again and the cell pellet is resuspended in 50 μL of D3 media. The cells are then plated in an 18 mm round cover previously coated by incubating one glass surface with 0.05 mg/mL Concanavalin A (Sigma-Aldrich C2272) in PBS 1x during 1 h, washed and covered with D3 media. The cells are then incubated for 40 min to allow cell adhesion and fixed in cold methanol for 3 min. The immunofluorescence is performed as in the "Immunofluorescence" section.

## Morphological zebrafish embryo analysis

Zebrafish embryos at 36 hpf were anesthetized with tricaine methane sulfonate (MS-222, Sigma-Aldrich) The morphological phenotype was evaluated on-site with an Olympus SZX16 scope equipped with an Olympus DP73 camera. Representative images were analyzed with ImageJ 2.3.0.

## TCGA data and aneuploidy scores

The XenaBrowser[51] was used to obtain publicly available data on pan-cancer normalized gene expression, methylation, and somatic mutations in patients from The Cancer Genome Atlas (TCGA). Aneuploidy scores were obtained directly from Supplementary Table 2 in ref. 23. Gene lengths were obtained from the latest human genome reference (GRCh38) made available through the object "ens.gene.ann.hg38" in the R package GeneBreak. From all cancer types available in TCGA, only those with at least 20 samples of solid healthy and primary tumor tissue in the gene expression matrix ($n=13$) were considered.

The full analysis pipeline is available at https://github.com/MiqG/publication_zadra_ttll11.

## Differential expression analysis of *TTLL11*

A Wilcoxon rank-sum test was performed for every type of cancer, to statistically assess the differences in log-normalized read counts of *TTLL11, TTLL2, TTLL4, TTLL5, TTLL6, TTLL7, TTLL9, TTLL11*, and *TTLL13* between primary tumors and unmatched healthy solid tissue samples. We performed the same procedure to compare the expression levels of TTLL11 and TTLL13 within tumor samples (Supplementary Fig. 6g).

## Association between *TTLL11* expression and sample aneuploidy

For each type of cancer, the Spearman correlation between log-normalized read counts of every gene and the corresponding sample aneuploidy score was computed. It was then assessed how likely it is to obtain the correlation coefficient of *TTLL11* with respect to the rest of the genes. The correlation coefficients were standardized, and the one-sided $P$ value of the correlation of *TTLL11* was computed with respect to the full distribution. Finally, to have an overview of the association between *TTLL11* expression and aneuploidy across cancers with respect to the rest of genes, the median correlation coefficient was computed across all cancer types and re-computed the $P$ value (as explained).

## Mutation frequency of *TTLL11*

For every type of cancer and annotated mutation effect, the mutation frequency per gene was computed and then divided by the corresponding gene length to obtain the frequency of mutations per kilobase. The same procedure was followed as explained above to obtain the one-sided $P$ value of the standardized mutation frequency of *TTLL11* per kilobase across cancer types and mutation effect with respect to the rest of the genes.

## Co-expression-based enrichment analysis

Across all cancer types, we computed the pairwise Spearman correlation coefficients between each TTLL glutamylase enzyme and the rest of the genes in healthy solid tissue and primary tumor samples,

separately. These co-expression signatures were then used to perform gene set enrichment analysis (GSEA) with the Gene Ontology (GO) biological processes and the ChEA[24]) datasets using the "gseGO" and "GSEA" functions, respectively, from the R package "clusterProfiler"[52]. For the GSEA based on known transcription factor targets from ChEA, we downloaded the gene lists for each transcription factor from the Harmonizome database [https://maayanlab.cloud/Harmonizome/dataset/CHEA + Transcription+Factor+Targets]. To identify the tumor-altered co-expression partners of each TTLL glutamylase, we subtracted the normalized enrichment scores (NES) of enriched (FDR < 0.05) gene sets in tumor signatures minus the solid tissue normal signatures.

### Differential expression analysis of TTLL11 upon overexpression of oncogenes

We downloaded the raw gene counts matrix and sample metadata from GSE185512; ref. 27. We then transformed the raw gene counts into transcripts per million (TPM) using the median gene lengths from the hg38 genome reference calculated with the python package GTFtools (Li, H.-D. GTFtools: a Python package for analyzing various modes of gene models. 263517 Preprint at https://doi.org/10.1101/263517 (2018). Their experiments contained the RNA sequencing profiles of 4 different cell lines (RPE1, MDAMB231, HCC1806, BT549) upon over-expression or not of three oncogenes (CCNE1, CDC25A, or MYC) for either 48 or 120 h with two replicates per time point and treatment. After log-transforming gene TPMs with $log2(TPM + 1)$, we normalized the TPMs of each condition (time point and treatment) with respect to the average expression in the corresponding negative control ("Empty" vector) by subtracting their log-TPMs.

Finally, we performed a Wilcoxon rank-sum test for every oncogene comparing the expression of TTLL11 upon treatment or not with doxycycline (i.e., overexpression or not the corresponding oncogene).

### Stratification of TCGA primary tumor samples based on the expression of CCNE1 and CDC25A

After observing that overexpressing CCNE1 and CDC25A increases the expression of TTLL11 in vitro, we stratified patients based on the expression of these genes to confirm this relationship in primary tumors as well. For each type of cancer, we considered the 10% of samples with the highest expression of either of the genes as "CCNE1_high or CDC25A high", and the 10% of samples with the lowest expression of both genes as "CCNE1_low & CDC25A_low". Subsequently, we performed a Wilcoxon rank-sum test for every cancer type comparing the expression of TTLL11 in patient samples of each primary tumor class versus unmatched healthy solid tissue.

### Differential methylation analysis of TTLL11

First, we obtained the coordinates for the 1500 bp upstream of TTLL11's transcription start site in the hg19 genome annotation, which corresponds to the coordinates of the methylation probes. Second, we used these coordinates to compute the median methylation of the probes found within the specified coordinates in each sample. Finally, we performed a Wilcoxon rank-sum test for every cancer type to compare the methylation levels of TTLL11 promoter between primary tumors and healthy solid tissue samples.

### Sequence sampling

Amino acid sequences were retrieved from Ensembl (www.ensembl.org). Annotated canonical sequences of human (GRCh38.p13) TTLLs and their orthologous were used for the following species: *Mus musculus* (GRCm39), *Danio rerio* (GRCz11), *Drosophila melanogaster* (BDGP6.32) and *Caenorhabditis elegans* (WBcel235). Transcripts used in the phylogenetic analysis are reported in Supplementary Table 5 (with assembled genome IDs of the different species in brackets).

### Alignment estimation and phylogeny using Bali-phy

Popular alignment tools fail to provide a reliable alignment due to the highly divergent protein sequence employed in this analysis[53]. Hence, Bali-phy v.3[54] was implemented to analyze the TTLL amino acid dataset; this program uses a model-based approach, so the alignment is calculated alongside the phylogenetics analysis using Bayesian inference. This method estimates the alignment with a likelihood averaged across several alignments[55]. This multiple alignment approach is superior for estimating alignment in highly divergent sequences than the most common alignment tool as MAFT and MUSCLE[56]. Moreover, it overcomes the alignment-guided tree that relies already on a tree topology for estimating the alignment[53]. The analysis was run three times to check for convergence among different runs. Parameters used were: alphabet amino acids -smodel gtr. Rates.gamma -imodel rs07[56].

### Software

See Supplementary Methods for a list of software used for data analysis.

### Reporting summary

Further information on research design is available in the Nature Portfolio Reporting Summary linked to this article.

## Data availability

All relevant data supporting the key findings of this study are available within the article and supplementary Information, and available from the corresponding author upon request. Previously published and publically available datasets used in the course of this work are listed in the relevant Methods sections and additionally in Supplementary Methods. Source data are provided with this paper.

## Code availability

The full bioinformatic analysis pipeline is available at https://github.com/MiqG/publication_zadra_ttll11 and https://doi.org/10.5281/zenodo.7298651.

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

## Acknowledgements

The work was supported by the European Union's Horizon 2020 research and innovation program under the Marie Skłodowska-Curie grant agreement No. 675737 to I.V., I.Z., and C.J.; the grants from the Spanish Ministry of Economy (MINECO) I + D grant BFU2012-37163 and BFU2015-68726-P to I.V.; C.J. was supported by the French National Research Agency (ANR) award ANR-17-CE13-0021 and Fondation pour la Recherche Medicale (FRM) grant DEQ20170336756. V.R. was supported by the Spanish Ministry of Economy (MINECO) I + D grant PID2020-117011GB-I00. We thank all the members of the Vernos group and the DiviDE ITN for the discussions and the CRG microscopy facility for technical support. We acknowledge the Spanish Ministry of Economy, Industry and Competitiveness (MEIC) to the EMBL partnership and support of the Spanish Ministry of Economy and Competitiveness, "Centro de Excelencia Severo Ochoa" as well as support of the CERCA Programme/Generalitat de Catalunya. The Genotype-Tissue Expression (GTEx) Project was supported by the Common Fund of the Office of the Director of the National Institutes of Health, and by NCI, NHGRI, NHLBI, NIDA, NIMH, and NINDS. The data used for the analyses described in this manuscript are in part based upon data obtained from the GTEx Portal on 05/03/2021. The results shown here are in part based upon data generated by the TCGA Research Network: https://www.cancer.gov/tcga.

## Author contributions

I.Z. performed all experiments in HeLa cells, and in RPE1 cells, and the processing of zebrafish embryos and analyzed the data. S.J. performed zebrafish microinjections and RT-PCR, generated the biallelic knockout, and analyzed cell size and apoptosis. M.A.-G. performed all the bioinformatic analysis. C.S.-M. analyzed spindle MT polyglutamylation levels in RPE1 and cancer cell lines. Z.C. tested the activity of the mutant TTLL11 E531G in HEK293 cell lysates and characterized lagging chromosome frequencies in RPE1 cells as well as several IFs. C.J. provided all the human TTLL clones. L.S. supervised M.A.-G. and C.S.-M. V.R. supervised S.J. and contributed to the experimental design. I.V. designed the research project and supervised directly I.Z. and Z.C. I.V., I.Z., and M.A. wrote the manuscript with contributions from all authors.

## Competing interests

The authors declare no competing interests.
