## [Peer Review File · Nature Communications]

Chromosome segregation fidelity requires microtubule polyglutamylation by the cancer downregulated enzyme TTLL11REVIEWER COMMENTS

Reviewer #1 (Remarks to the Author):

In this manuscript, the authors found that one of the polyglutamylases, TTLL11, specifically localizes at mitotic spindles which is important for polyglutamylation of mitotic spindles and chromosome segregation fidelity. Comparisons among different kinds of normal tissues and cancer tissues showed that TTLL11 is generally downregulated in human tumors which frequently exhibit aneuploidy. Such correlation may imply a molecular link between microtubule polyglutamylation, chromosome segregation, and tumorigenesis. Although authors have tested the possible mechanisms as of why TTLL11 downregulates in tumors, the findings in this manuscript only demonstrated that mutation rate of TTLL11 gene in tumor patients is not the reason. Moreover, authors also found that SAC, bipolar spindle assembly, or kinetochore attachment are not the factors which lead to chromosome missegregation in hypoglutamylated mitotic spindles. The evidence of TTLL11 downregulation in tumors and hypoglutamylated mitotic spindles cause aberrant chromosome segregation needs to be provided for considering the publication in Nature Communications.

Major points:

1. Since the roles of tyrosination/detyrosination in metaphase chromosome movement have been reported, it is necessary to confirm whether TTLL11 modulates chromosome segregation is solely dependent on polyglutamylation or is also dependent on tyrosination/detyrosination. To address this, the level of tyrosination/detyrosination in TTLL11-overexpressed cells needs to be evaluated.
2. Besides the metaphase, the distribution of GFP-TTLL11 in interphase and telophase needs to be evaluated. It's plausible that TTLL11 may have also contributed to the polyglutamylation of intercellular bridges which in turn recruits spastin to ensure proper abscission of intercellular bridges. The micronuclei and unfaithful chromosome segregation may have resulted from aberrant abscission of intercellular bridges.
3. The enzyme activity of TTLL11(E466E) needs to be analyzed, at least, in the experiment in supplementary Fig. S2a.
4. The detailed mechanisms of TTLL11 downregulation in tumor cells need to be explored since the mutation rate of TTLL11 gene can not explain this key concept in the manuscript.
5. Whether expression of TTLL11 can rescue the aberrant chromosome segregation in different tumor cells need to be confirmed.
6. The detailed molecular mechanism of how hypoglutamylated mitotic spindles leads to chromosome missegregation needs to be explored.
7. Why is GFP-zfTTLL11 localized in metaphase plate in Figure 2c?

Minor points:

1. Remove "the majority of" in the first sentence on page#4.
2. The description of Supplementary Movie S3, and S6 in result section is missing.
3. The labeling of x-axis in figure 4 needs to be corrected.
4. Remove the description about error bars in the legend of figure 1b.
5. "eef1A" is a typo in the legend of figure 2

Reviewer #2 (Remarks to the Author):

Zadra and colleagues investigate the molecular mechanism and function of tubulin polyglutamylation of mitotic spindle microtubules using human HeLa cells and zebrafish as model systems. By expressing GFP-tagged enzymes involved in polyglutamylation, combined with gene expression analysis, the authors identify TTLL11 as the main enzyme accounting for spindle microtubule polyglutamylation (long chains) in HeLa cells. To investigate function, the authors perform siRNA and Morpholino analyses in HeLa cells and zebrafish blastula stage embryos. Specificity is confirmed with appropriate

rescue experiments, including catalytic dead version of TLL1. The authors report a significant increase in chromosome segregation errors in both systems and report that TLL1 is often downregulated in human cancers and their level of aneuploidy. Investigation of the impact of TLL1 depletion on spindle microtubule dynamics, suggest that TLL1 depletion hyperstabilizes spindle microtubules, including k-fibers, providing a mechanistic explanation for the role of TLL1 in mitotic fidelity. Overall, this study investigates a long-standing question in mitosis and the results, if supported by additional controls and extended to a level of deeper mechanistic understanding, will be of interest to a wide readership.

Major issues:

1- Tubulin deetyrosination has been recently implicated in error-correction and mitotic fidelity in human cells (Ferreira et al., JCB, 2020). This should be acknowledged in the present manuscript and the impact of TLL1 depletion on tubulin deetyrosination overall (western blot) and of spindle microtubules (immunofluorescence) investigated, as this may offer an alternative explanation for the increase in mitotic errors upon TLL1 depletion.

2- The choice of HeLa cells as "normal cells" to investigate mitotic fidelity is unfortunate as these are highly transformed cells. The key findings, such as the role of TLL1 in spindle microtubule polyglutamylation and the consequent increase in mitotic errors, should be confirmed in non-transformed human cells, such as RPE1.

3- The IF data on TLL localization on spindle microtubules is not totally convincing and should be improved. First, the authors should compare with GFP alone signal on spindle microtubules. Even better control, the authors should show the localization of the other TLLs that did not enrich on spindle microtubules for comparison and provide full details on image acquisition and processing (not provided in the methods). Quantification of spindle signal relative to cytosolic levels should be provided for all conditions and proper stats.

4- The authors refer to TLL as THE enzyme that accounts for spindle microtubule polyglutamylation. Yet, TLL1 depletion by siRNA causes less than 50% reduction in polyE levels. Is this due to depletion efficiency? The authors can only indirectly infer for this using the transfected cells with GFP-TLL1, and overall levels of polyE seem unaffected by TLL1 depletion (sup. fig. 2). On the other hand, GFP-TLL1 overexpression causes a massive increase in polyE. Is this associated with any mitotic phenotype? This section requires attention by the authors and an explanation for the remaining polyE (on spindles) after TLL1 depletion must be provided and the conclusions toned down accordingly.

5- Related point, TLL1 and polyE localization in zebrafish embryos is very different from the one in HeLa cells and appears to decorate the chromosomes. This should be explained and discussed.

6- Results, page 6, the authors first indicate that TLL1 depletion in HeLa cells causes no problem in segregation efficiency and in the next sentence say that mitotic fidelity was compromised. This requires clarification.

7- Results, pages 7 and 8. The authors indeed show many examples of lagging chromosomes in anaphase in zebrafish embryos, but from the movies provided, only in one case lagging chromosomes resulted in micronuclei, otherwise they mostly correct and reintegrate the main nuclei. Since the origin of most micronuclei was not traced in these experiments, the authors should use caution in drawing conclusions about micronuclei origin. If any conclusion is drawn, proper quantification and tracing must be provided. Related to this, in figure 1e the IF images provided show a chromatin bridge (right) and a lagging (left). How were these distinguished in the quantifications?

8- Results, page 10, the authors state that "The time required for cells to enter anaphase upon STLC washout can be used as a readout for the presence of a functional error correction mechanism" and the conclusion thereof is not correct, as many incorrect attachments satisfy the SAC and therefore do not necessarily cause a delay in anaphase onset. A more direct readout would be frequency of errors in anaphase and the likelihood to form micronuclei.

9- Results, page 11, although a slight reduction in flux rates is somewhat indicative of alterations in microtubule dynamics, several other factors and processes (such as motor protein-mediated microtubule sliding) are involved in flux (see e.g. Steblyanko et al., EMBO J, 2020). A more direct measure of microtubule stability would be to calculate the half-life of the different spindle microtubule

populations from the photoactivation experiments by fitting a double exponential curve to the fluorescence dissipation curves. Indeed, from the data provided, it seems that the photoactivation mark remains stronger over time after TLL11 depletion.

10- It remains unclear whether TLL11 affects microtubule dynamics directly or indirectly. The authors briefly allude to this in the discussion, but this level of mechanistic insight would be expected to merit publication in Nature Communications. Obvious candidate proteins that are regulated by polyglutamylation, such as spastin (also implicated in spindle flux), should be tested, in addition to ruling out indirect effects over tubulin deetyrosination and other well-established targets such as MCAK. Minor issues:

1- Introduction page 3: the statement about the SAC is incorrect, there are attachments that satisfy the SAC that are incorrect (merotelic). The origin of microtubules at kinetochores is also hotly debated and evidence that they (all) emanate from centrosomal microtubules is lacking. Sentence should be re-phrased.

2- Top of page 4: it seems that there is a strikethrough in "the majority of". Please correct.

Reviewer #3 (Remarks to the Author):

The authors show that TLL11 is responsible for polyglutamylation of the mitotic spindle in HeLa cells. To assess function of spindle polyglutamylation, they deplete TLL11 from HeLa cells and observe chromosome segregation defects. In parallel, they show that TLL11 depletion in zebrafish embryos leads to the formation of micronuclei. The authors then analyzed the TCGA database and found a systematic downregulation of TLL11 in cancer cells and they correlate this with aneuploidy. Finally, the authors show that spindle MTs are more stable in the absence of polyglutamylation. They conclude that polyglutamylation tunes MT dynamics to ensure faithful chromosome segregation.

Although understanding the role of spindle polyglutamylation during mitosis is a very interesting question, the data provided in this manuscript is not very convincing. The conclusions result from correlations between data obtained in different systems (HeLa cells, zebrafish embryos) and cancer cell databases without linking them together.

Major concerns:

Introduction Page 4: "During mitosis, spindle MTs are modified with numerous PTMs, including acetylation, deetyrosination and polyglutamylation. However, still little is known about their roles in cell division, except for deetyrosination, which guides metaphase chromosome congression via the motor protein CENP-E."

While it is true that the function of polyglutamylation in mitosis remains unknown, a recent study highlights a role for MT acetylation in maintaining spindle bipolarity in epithelial cells (Rasamizafy et al. Cells 2021).

Results Page 4: Among the different TLL proteins, only TLL11 and TLL13 localize to the spindle when overexpressed. Wouldn't it be expected to have at least one of the initiating polyglutamylases? How do the authors explain this?

Page 7: The authors show that zTLL11-MO injected embryos have severe developmental defects and conclude that this suggests "a pivotal role for TLL11-dependent spindle MT polyglutamylation". To validate this conclusion, the authors should show that polyglutamylation is decreased or abolished on spindle MTs following TLL11-MOs. Although polyglutamylation of spindle MT likely contributes to the observed phenotype, the authors cannot exclude a contribution of the polyglutamylation of other MTs, such as those from the nodal cilia known to play critical roles in early embryo patterning. Did the authors look whether the polyglutamylation levels in the zebrafish cilia are affected by TLL11-MOs?

Page 7: The authors observe micronuclei in zebrafish embryos. Did they detect similar micronuclei in

HeLa cells after TLL11 siRNA treatment or did cells finally manage to divide properly? It would be interesting to know the consequences of the observed increase in lagging chromosomes in HeLa cells.

Page 9: Using TCGA database, the authors show a systematic downregulation of TLL11 in cancer cells. Does this mean that spindle polyglutamylation is systematically downregulated in cancer cells? The authors use cancer cells (HeLa cells) for their study. Wouldn't it be more appropriate to use "normal" cells to address the role of spindle polyglutamylation? Do HeLa cells have low levels of polyglutamylation as compared to "normal" cells, for example how does their polyglutamylation level compare with the one of zebrafish embryo cells? The link between spindle polyglutamylation and cancer would be more convincing if authors had showed a systematic decrease in polyglutamylation in cancer cells.

Page 11: The authors found that siTLL11-depleted cells have more stable spindles. Do they observe changes in the levels of MT acetylation or detyrosination two PTMs known to increase with MT stability? Could these changes contribute to the observed phenotype?

Minor points:

Top page 4 : remove « the majority of ».

Page 5: reference to figure S3b is misleading as it follows the sentence " the polyE signal was specifically and significantly reduced in spindles assembled in siTLL1 cells as compared to control cells". Figure S3b shows that polyE is not reduced in interphase cells depleted of TLL11. It might be more comprehensive for the readers to state this point more clearly. This would also help the readers to understand why in figure S2a, the basal polyglutamylation level of HeLa cells is not reduced after siTLL11 when examined by immunoblotting.

Movie S4: While H2A is labelled with mCherry and TLL11 with GFP, the colors are inverted in the movie. Same is true for the tubulin/H2B in Movie S3.

Top Page 11: TLL111 instead of TLL11 and K-fibres instead of K-fibers.

Reviewer #4 (Remarks to the Author):

Reviewer comments

In the manuscript, the authors identified that the enzyme TLL11 regulates polyglutamylation and microtubule dynamics required for chromosomal stability during mitosis in vitro and in vivo. This enzyme polyglutamates the spindle microtubules in mitosis, which support microtubule dynamics and error-free chromosomal segregation. By using HeLa cell and zebrafish, the authors showed that decreasing TLL11 expression level increased the probability of abnormal chromosomal segregation. Additionally, they found that TLL11 is downregulated in human tumors, using a human cancer data base. Overall, the authors provide evidences for a new mechanism of spindle control by one specific post-translational modification of tubulin. Hence these results may be of interest to a broad audience. I included below major and minor points that will improve the quality of the manuscript and recommend them to be addressed prior to a possible publication.

Major comments

1. Expression of TLL11

1-1. The authors overexpressed fluorescence tagged TLLs as a strategy to visualize their localization.

I understand that the absence of antibodies limit the analysis of the endogenous expression pattern. Nevertheless, I wonder if overexpressing TTLLs could affect their endogenous location due to for instance increased activity and tubulin glutamylation (as shown e.g. Fig S2A). One way to address this would be to provide the data for the localization and impact on polyglutamylation of the other overexpressed TTLLs described in the text, and/or to titrate the expression levels of GFP-TTLL.

1-2. The GFP-TTLL11 expression in HeLa cells is observed in the entire cell in addition to the spindle (Figure 1A). On the other hand, the GFP-zTTLL11 expression in zebrafish blastomere seems to be more restricted in the spindle (Fig 2C). How this difference can be explained? Is it due to the overexpression strategy (e.g. different TTLL11 protein quantity)?

2. Strategies to study the role of TTLL11

2-1. The authors used HeLa cells for the loss-of-function study of TTLL11. However, they also mentioned in the manuscript that the levels of TTLL11 is reduced in most tumors. I am wondering whether reducing TTLL11 in HeLa cells is the best approach to test the role of TTLL11 in microtubule dynamics and mitosis. I recommend the author to test their findings in other cells, if possible not cancerous cells.

2-2. In the zebrafish model, the authors only used MO strategy to knockdown TTLL11. Even though they showed mRNA rescue experiment, the MO strategy is still arguable in the zebrafish society. Currently, F0 biallelic knockout by CRISPR/Cas9 is widely used in zebrafish. Hence, I suggest that they should show the identical phenotype via CRISPR/Cas9 system.

3. Analysis of knock-down phenotypes.

3-1. TTLL11 knock-down affect the levels of polyE tubulin in the spindle, but on the WB (Fig S2A), there is no effect on total polyE tubulin. In a similar line of thought, the authors do not show the impact of TTLL11 overexpression on polyE tubulin by staining. Given the WB shown in S2A, I would expect the increase of polyE tubulin to occur broadly in the cell. Taking these two observations into consideration, could the authors clarify how TTLL11 could specifically regulate the polyE on the spindle and not elsewhere?

3-2. The authors showed abnormal chromosome segregation and severe developmental defects in zebrafish by downregulating TTLL11. It seems that the abnormal segregation generates aneuploidy and chromosome instability. However, the authors did not show that directly. The authors should provide a direct evidence that TTLL11 downregulation induces aneuploidy.

3-3. In the current manuscript, the authors did not explain sufficiently why the zebrafish have such strong developmental defects. Are these due to e.g., increased apoptosis or reduced proliferation? It is particularly interesting that zebrafish show such a severe phenotype given that HeLa cells do not show any changes in their mitotic index. I would recommend the authors to describe and discuss better how similar subcellular defects (genome instability) in the zebrafish embryos and HeLa cell line show different phenotypes (proliferation vs developmental defect/apoptosis).

3-4. Currently the authors do not show if zebrafish embryos injected with the TTLL11 MO show reduced polyE glutamylation in the spindle. Adding these results is crucial to further validate that the observations in downregulated HeLa cells hold true in the zebrafish too.

4. Additional discussion

The authors mainly focused the localization and function of TTLL11 in mitosis. However, they still emphasize polyglutamylation in the chromosomal fidelity in the manuscript. I noticed that TTLL13 is also localized in the spindle at mitosis, and it is increased in several tumors in contrast to TTLL11. It

might be worthy to describe about this additional player for glutamylation in the discussion part.

Minor comments

1. The authors did not describe what TTLL11 stands for. Please provide the full name before using abbreviation.

2. The supplementary movie files are not in order. For example, the authors described the movie S5 before using the movie S2. Please arrange the name of files in orders for readers' convenience.

3. There are a few typos, abnormal marks or reference not added. Please fix them.

- Page 4, line 1: the majority of
- Page 11, line 2: TTLL111
- Page 11, line 7: (Warren et al, 2020).
- Page 32, line 1: .p, 0.01,

4. Please avoid the use of red and green colors in your merge confocal image for color-blind accessibility.

Point-by-point responses to the reviewers' comments

We first want to thank all the reviewers for the numerous concerns and suggestions that we have addressed with new experimental data and have helped us improve our manuscript.

Reviewer #1 (Remarks to the Author):

In this manuscript, the authors found that one of the polyglutamylases, TTLL11, specifically localizes at mitotic spindles which is important for polyglutamylation of mitotic spindles and chromosome segregation fidelity. Comparisons among different kinds of normal tissues and cancer tissues showed that TTLL11 is generally downregulated in human tumors which frequently exhibit aneuploidy. Such correlation may imply a molecular link between microtubule polyglutamylation, chromosome segregation, and tumorigenesis. Although authors have tested the possible mechanisms as of why TTLL11 downregulates in tumors, the findings in this manuscript only demonstrated that mutation rate of TTLL11 gene in tumor patients is not the reason. Moreover, authors also found that SAC, bipolar spindle assembly, or kinetochore attachment are not the factors which lead to chromosome missegregation in hypoglutamylated mitotic spindles. The evidence of TTLL11 downregulation in tumors and hypoglutamylated mitotic spindles cause aberrant chromosome segregation needs to be provided for considering the publication in Nature Communications.

Major points:

1. Since the roles of tyrosination/detyrosination in metaphase chromosome movement have been reported, it is necessary to confirm whether TTLL11 modulates chromosome segregation is solely dependent on polyglutamylation or is also dependent on tyrosination/detyrosination. To address this, the level of tyrosination/detyrosination in TTLL11-overexpressed cells needs to be evaluated.

We thank this reviewer for pointing out this very important issue. We have now addressed it directly by quantifying the levels of tyrosinated and detyrosinated MTs by IF in metaphase spindles assembled in control and TTLL11 silenced Hela cells. We found no significant changes in the levels of spindle MT tyrosination or detyrosination in control and silenced cells (Fig. 1c, page 6 in results section and page 21 in the methods section). This strongly suggests that the tubulin tyrosination-detyrosination cycle is not altered in mitotic TTLL11 silenced cells.

In addition, we also checked whether for any change in MT acetylation levels in silenced cells. We did not find any difference between control and siTTLL11 spindles (Fig.1c).

Overall, we conclude that TTLL11 silencing specifically impairs MT polyglutamylation without interfering with other PTMs.

2. Besides the metaphase, the distribution of GFP-TTLL11 in interphase and

telophase needs to be evaluated. It's plausible that TTLL11 may have also contributed to the polyglutamylation of intercellular bridges which in turn recruits spastin to ensure proper abscission of intercellular bridges. The micronuclei and unfaithful chromosome segregation may have resulted from aberrant abscission of intercellular bridges.

Overexpressed GFP-TTLL11 localizes to the nucleus and the cytoplasm in interphase cells. Interphase MTs are glutamylated as revealed by their labelling with the antibody GT335 that recognizes the branching glutamate (Wolff et al., 1992; Van Dijk et al, 2007 and our own data shown in Supp. Fig. 2f, g), but they are not or very weakly labelled with the polyE antibody indicating that the branching glutamate chains are very short (1 to 2 glutamates). As previously described in van Dijk et al (2007), overexpressed GFP-TTLL11 in the cytoplasm drives the aberrant polyglutamylation of the interphase MTs (Supp. Fig.2 d, e) and the reorganization of the MT network. We also observed an increase in cell death.

Following the reviewer's suggestion, we further evaluated a putative role of TTLL11 during abscission. We did not observe phenotypes consistent with a defect in abscission may be compromised, such as bi- or multinucleated cells in tissue culture cells or zebrafish embryos in vivo in the absence of TTLL11. Instead, we consistently found an increase of lagging chromosomes in silenced Hela (Fig.1e) and hTERT-RPE1 cells (Supp. Fig.3c) as well as in morpholino injected (Fig.3a, b) and in the newly generated biallelic knockouts of zebrafish embryos (Supp. Fig. 5c, d). We could also trace the fate of lagging chromosomes in zebrafish embryos to micronuclei (Fig. 3c), suggesting that this is the mechanism behind micronuclei formation.

3. The enzyme activity of TTLL11(E466E) needs to be analyzed, at least, in the experiment in supplementary Fig. S2a.

It was previously shown that all TTLL family enzymes share a conserved E residue in their catalytic domain that is essential for enzymatic activity (Van Dijk et al, 2007). Sequence alignment of the TTLL11 proteins from different organisms showed that the essential E438 (in murine TTLL11) corresponds to E531 in human TTLL11 and to E466 in zebrafish (Supp. Fig. 4b).

For a further direct confirmation, we performed an activity test for the human TTLL11 point mutant enzyme E531G (Supp. Fig.4c). We transfected HEK293 cells with constructs expressing GFP-TTLL11 or TTLL11-YFP or TTLL11-YFP-E531G. In agreement with previous reports, Western blot analysis of the cell lysates showed that TTLL11 overexpression drives a substantial increase of tubulin polyglutamylation (Van Dijk et al 2007). Note that cells were not synchronized and therefore most of them were in interphase, therefore the signal for polyglutamylated MTs (with the anti PolyE antibody) is extremely weak in control cells but strong in cells overexpressing TTLL11. Instead, overexpression of TTLL11-E4531G does not drive any increase in tubulin polyglutamylation. These data demonstrate that introducing the point mutation E531G abolishes TTLL11 polyglutamylase activity (Supp. Fig.4c).

Furthermore, in agreement with these results we found that the catalytic inactive point mutant zfTTLL11(E466G) does not rescue TTLL11 morphants developmental phenotypes (Fig. 2e) nor the chromosome segregation defects (Fig.3b).

4. The detailed mechanisms of TTLL11 downregulation in tumor cells need to be explored since the mutation rate of TTLL11 gene cannot explain this key concept in the manuscript.

We have now extended the bioinformatic analysis to explore the mechanism that may account for the downregulation of TTLL11 in human cancer. The new data are now included in a new section in Results (pages 12-14), in Fig. 6 and in Supplementary Fig.7 and 8.

First, we obtained genome-wide co-expression profiles for each (poly)glutamylase TTLL and the rest of the human genes in healthy and tumour tissues (Supp.Fig.8a). A clear correlation was found for all of them except for TTLL11 that showed a distinct pattern with the lowest correspondence of co-expression profiles in tumours versus healthy samples. Therefore, TTLL11 is the TTLL polyglutamylase with the strongest tumour-specific co-expression signature in cancer.

In addition, we found that the consistent downregulation of TTLL11 across tumours can be explained by the overexpression of the oncogenes CCNE1 or CDC25A as part of a tumour-specific transcriptional program in cell lines and in human tumours (Fig.6).

Altogether, we provide now more compelling evidences for the specific downregulation of TTLL11 in cancer.

5. Whether expression of TTLL11 can rescue the aberrant chromosome segregation in different tumor cells need to be confirmed.

Tumour cells have extensively altered gene expression programs and present a wide variety of chromosome segregation errors that result in aneuploidy and can promote CIN. We believe that in such highly altered background, MT polyglutamylation (and TTLL11 expression) is most probably not the sole factor needed to rescue chromosome segregation fidelity. In addition, we found that overexpressing TTLL11 in Hela cells results in high levels of cell death. We interpret this result as the consequence of the resulting aberrant interphase MT polyglutamylation and probably as a consequence the reorganization of the interphase MT network. We therefore believe that exogenous expression of TTLL11 is unlikely to rescue chromosome segregation errors in cancer cells.

However, we have shown that expression of TTLL11 but not the catalytic dead enzyme rescues chromosome segregation errors (and micronuclei formation) in vivo using the early zebrafish embryo as a model system (Fig. 2e and 3b).

6. The detailed molecular mechanism of how hypoglutamylated mitotic spindles leads to chromosome missegregation needs to be explored.

Our data suggest that the mechanism involves changes in spindle MT dynamics as we detected an increase of spindle size, an increased resistance to cold induced MT depolymerization in mitotic cells and a reduced speed of the spindle flux. These changes are relatively mild albeit significant. These changes do not interfere with bipolar spindle assembly and with the general spindle function in chromosome segregation. However, they trigger errors in chromosome segregation both in HeLa and hTERT-RPE1 cells and in zebrafish early embryos. This is fully consistent with previous reports indicating that a mild increase in MT stability triggers chromosome segregation errors and with the reported 'hyper' stabilization of the MTs in cancer cells.

We further explored the localization of a number of spindle proteins (p150, Katanin, hKlp2, Eg5 and MCAK) but did not detect significant changes in silenced cells (see **figures 1 and 2** attached at the end for the reviewers). These observations strongly support the hypothesis that the reduction of MT polyglutamylation of the spindle MTs will result in relatively small changes of MT binding affinities and/or activity of a number of associated proteins. These small changes collectively alter MT dynamics and compromise the fidelity of chromosome segregation.

7. Why is GFP-zfTLL11 localized in metaphase plate in Figure 2c?

We can indeed visualize a faint localization of GFP-zfTLL11 on the metaphase plate during live imaging of dividing cells in zebrafish embryos although not consistently. We do not know why. One possibility is that it depends on the levels of overexpressed protein and the nucleo-cytoplasmic protein ratio. GFP-zfTLL11 may associate transiently with the chromosomes during the initial phases of cell division. An alternative hypothesis is that TLL11 may associate at sites of chromosome-MT attachments.

Although we did not observe any chromosome localization in HeLa cells overexpressing TLL11, this localization could be missed as it appears to occur only transiently for a short time window that cannot be captured in fixed cells.

Minor points:

1. Remove “the majority of” in the first sentence on page#4.

Thank you for noticing this mistake in the text. We removed it.

2. The description of Supplementary Movie S3, and S6 in result section is missing.

We carefully revised the order of the movies and now properly reference them all in the results section.

3. The labeling of x-axis in figure 4 needs to be corrected.

We apologize, a rearrangement of the figure happened during the conversion to pdf and we failed to notice it. We have now carefully checked that the conversion to pdf is correct.

4. Remove the description about error bars in the legend of figure 1b.

We removed this. Thank you for noticing this mistake.

5. “eef1A” is a typo in the legend of figure 2

Thank you, we have corrected this mistake to the right nomenclature: eef1a1 (eukaryotic translation elongation factor 1 alpha 1)

Reviewer #2 (Remarks to the Author):

Zadra and colleagues investigate the molecular mechanism and function of tubulin polyglutamylation of mitotic spindle microtubules using human HeLa cells and zebrafish as model systems. By expressing GFP-tagged enzymes involved in polyglutamylation, combined with gene expression analysis, the authors identify TTLL11 as the main enzyme accounting for spindle microtubule polyglutamylation (long chains) in HeLa cells. To investigate function, the authors perform siRNA and Morpholino analyses in HeLa cells and zebrafish blastula stage embryos. Specificity is confirmed with appropriate rescue experiments, including catalytic dead version of TTLL11. The authors report a significant increase in chromosome segregation errors in both systems and report that TTLL11 is often downregulated in human cancers and their level of aneuploidy. Investigation of the impact of TTLL11 depletion on spindle microtubule dynamics, suggest that TTLL11 depletion hyperstabilizes spindle microtubules, including k-fibers, providing a mechanistic explanation for the role of TTLL11 in mitotic fidelity. Overall, this study investigates a long-standing question in mitosis and the results, if supported by additional controls and extended to a level of deeper mechanistic understanding, will be of interest to a wide readership.

Major issues:

1- Tubulin detyrosination has been recently implicated in error-correction and mitotic fidelity in human cells (Ferreira et al., JCB, 2020). This should be acknowledged in the present manuscript and the impact of TTLL11 depletion on tubulin detyrosination overall (western blot) and of spindle microtubules (immunofluorescence) investigated, as this may offer an alternative explanation for the increase in mitotic errors upon TTLL11 depletion.

We thank this reviewer for raising this very important point. We have added the reference in the introduction (page 5) and we have addressed the potential impact of TTLL11 silencing on the levels of tubulin detyrosination directly by IF. The quantification of the normalized signals for tyrosinated and detyrosinated MTs in spindles assembled in control and TTLL11 silenced Hela cells showed no significant changes between the two conditions (Fig. 1c, page 6 in results section and page 21 in the methods section). This strongly suggests that TTLL11 silencing has no impact on the levels of tubulin detyrosination in the spindle.

In addition, we also checked whether for any change in MT acetylation levels in silenced cells. We did not find any difference between control and siTTLL11 spindles (Fig.1c).

Overall, we conclude that TTLL11 silencing specifically impairs MT polyglutamylation without interfering with other PTMs.

2- The choice of HeLa cells as “normal cells” to investigate mitotic fidelity is unfortunate as these are highly transformed cells. The key findings, such as the role of TTLL11 in spindle microtubule polyglutamylation and the consequent increase in mitotic errors, should be confirmed in non-transformed human cells, such as RPE1.

We fully acknowledge this recommendation. We have now performed TTLL11 silencing experiments in the non-transformed human hTERT-RPE1 cells (Fig.5, Fig.7b, Supp. Fig 3). The results summarized below are fully consistent with those we obtained in HeLa cells.

Indeed, we found that TTLL11 silenced hTERT-RPE1 cells show:

- a decrease of the spindle MT polyglutamylation levels (Suppl. Fig.3a-b and Fig. 5)
- an increase of the spindle size (Fig.7b)
- an increased rate of lagging chromosomes (Supp. Fig.3c).

3- The IF data on TTLL localization on spindle microtubules is not totally convincing and should be improved. First, the authors should compare with GFP alone signal on spindle microtubules. Even better control, the authors should show the localization of the other TTLLs that did not enrich on spindle microtubules for comparison and provide full details on image acquisition and processing (not provided in the methods). Quantification of spindle signal relative to cytosolic levels should be provided for all conditions and proper stats.

The only localization data we could obtain was through overexpression of fluorescently tagged TTLLs because there are no antibodies to assess the localization of the endogenous proteins. Our aim was to identify putative candidate(s) that could drive the polyglutamylation of the spindle MTs through a qualitative analysis. We now provide as requested, the IF images from our screen showing metaphase cells overexpressing each of the 8 glutamylases in the TTLL family tagged with GFP (Supplementary Fig. 1a). These images show that only overexpressed fluorescently tagged TTLL11 and TTLL13 localize to the spindle. Since it was difficult to observe many metaphase cells expressing the tagged proteins, the results are qualitative rather than quantitative. We only used them as a starting point to select the candidate(s) enzyme and conduct validation studies.

We have also updated the Methods section and included the information about data acquisition and processing (pages 20-22).

4- The authors refer to TTLL1 as THE enzyme that accounts for spindle microtubule polyglutamylation. Yet, TTLL11 depletion by siRNA causes less than 50% reduction in polyE levels. Is this due to depletion efficiency? The authors can only indirectly infer for this using the transfected cells with GFP-TTLL1, and overall levels of polyE seem unaffected by TTLL11 depletion (sup. fig. 2). On the other hand, GFP-TTLL1 overexpression causes a massive increase in polyE. Is this associated with any mitotic phenotype? This section requires attention by the authors and an explanation for the remaining polyE (on spindles) after TTLL11 depletion must be provided and the conclusions toned down accordingly.

We had to check the efficiency of the siRNA by RT-PCR to monitor the decrease in TTLL11 mRNA because of the lack of antibodies to monitor the endogenous protein. We therefore do not know how much protein may still remain in the silenced cells and whether this may be enough to still drive a small level of spindle MT polyglutamylation. However, the reduction of the PolyE levels upon TTLL11 silencing in the spindle is significant and consistent. Moreover, we have now confirmed these data *in vivo* in morphant zebrafish embryos with reduced TTLL11 levels (Fig. 2f, g) and in TTLL11 silenced RPE1 cells (Fig. 5a and Supp.Fig.3a).

TTLL11 silencing indeed does not change the basal PolyE signal detected by western blot as shown in Supp Fig 2. Here we used non-synchronized cells that are therefore mostly in interphase. As previously described (Van Dijk et al, 2007), interphase MTs have a very low basal level of polyglutamylation as detected by IF (see Supp. Fig. 2d) and western blot analysis (see Supplementary Fig.2a). In the same blot (Supp. Fig.2a) we show that overexpression of GFP-TTLL11 causes an increase of interphase MT polyglutamylation, consistently with previous reports (Van Dijk et al, 2007). It means that when the enzyme is in the cytoplasm due to protein overexpression, it can polyglutamate the interphase MTs. Concerning phenotypes, GFP-TTLL11 overexpression causes a dramatic reorganization of the interphase MTs (that are now polyglutamylated) and increased cell death.

We could not assess whether there is any mitotic phenotype associated with TTLL11 overexpression, because very few overexpressing cells were actually in mitosis. Moreover, it would be difficult to dissociate a putative phenotype resulting from the specific 'over' glutamylation of the spindle MTs from a phenotype arising from the aberrant polyglutamylation of the interphase MTs prior to entering mitosis.

We have modified the abstract and main text (page 2 and 6) to refer to TTLL11 as **an** enzyme that polyglutamylates the spindle MTs.

5- Related point, TTLL1 and polyE localization in zebrafish embryos is very different from the one in HeLa cells and appears to decorate the chromosomes. This should be explained and discussed.

We monitored the localization of GFP-TTLL11 in fixed HeLa cells and in live zebrafish embryos. In both cases we could visualize the fluorescent protein on the metaphase spindle.

We can indeed also visualize a faint localization of GFP-zfTLL11 on the metaphase plate during live imaging of dividing cells in zebrafish embryos although not consistently. We do not know why. One possibility is that it depends on the levels of overexpressed protein and the nucleo-cytoplasmic protein ratio. GFP-zfTLL11 may associate transiently with the chromosomes during the initial phases of cell division. An alternative hypothesis is that TLL11 may associate at sites of chromosome-MT attachments.

Although we did not observe any chromosome localization in HeLa cells overexpressing TLL11, this localization could be missed as it appears to occur only transiently for a short time window that cannot be captured in fixed cells.

6- Results, page 6, the authors first indicate that TLL11 depletion in HeLa cells causes no problem in segregation efficiency and in the next sentence say that mitotic fidelity was compromised. This requires clarification.

Our data show that TLL11 silencing does not interfere with spindle assembly nor does it cause a delay in chromosome segregation in HeLa cells. To avoid confusion we changed the sentence in the results section page 6 that previously read '.....aligned and segregated chromosomes with a similar **efficiency**.' to '... aligned and segregated chromosomes with a similar **timing**.'

However, when we monitored anaphase we noticed an increase in lagging chromosomes. Therefore, we concluded that chromosome segregation fidelity is compromised.

7- Results, pages 7 and 8. The authors indeed show many examples of lagging chromosomes in anaphase in zebrafish embryos, but from the movies provided, only in one case lagging chromosomes resulted in micronuclei, otherwise they mostly correct and reintegrate the main nuclei. Since the origin of most micronuclei was not traced in these experiments, the authors should use caution in drawing conclusions about micronuclei origin. If any conclusion is drawn, proper quantification and tracing must be provided. Related to this, in figure 1e the IF images provided show a chromatin bridge (right) and a lagging (left). How were these distinguished in the quantifications?

To answer this concern, we analysed again carefully the movies of live zebrafish embryos to monitor the fate of lagging chromosomes. We found examples of at least one dividing cell with a lagging chromosome forming a micronucleus at the end of cell division in all the morphant embryos examined within a specific field of view capturing a small tissue section. This suggests that there are most probably more examples of these type of events per morphant embryos. These data are now shown in Fig. 3e.

We fully acknowledge the comment on the illustration of anaphase chromosome segregation defects in Fig.1e. We have modified Fig. 1e and its legend accordingly, as well as the main text to describe 'chromosome segregation defects including lagging chromosomes' (Results section, page 6 and pages 7, 8, 9).

8- Results, page 10, the authors state that “The time required for cells to enter anaphase upon STLC washout can be used as a readout for the presence of a functional error correction mechanism” and the conclusion thereof is not correct, as many incorrect attachments satisfy the SAC and therefore do not necessarily cause a delay in anaphase onset. A more direct readout would be frequency of errors in anaphase and the likelihood to form micronuclei.

The only incorrect attachments that are not detected by the SAC are merotelic attachments. All the others need to be corrected to satisfy the SAC before cells enter anaphase.

STLC treated cells arrest in mitosis with monopolar spindles that accumulate syntelic KT–MT attachments with both sister kinetochores attached to the same pole (or centrosome) (Lampson et al, 2004; Dudka et al, 2018). The time required for cells to correct these syntelic attachments and progress to anaphase after STLC washout is therefore a read out of the Aurora-B-dependent error correction efficiency (Lampson et al, 2004). We therefore used this validated assay to test for the functionality of the error correction mechanism in TLL11 silenced cells.

The error correction mechanism functions on all type of erroneous attachments including merotelic attachments. The SAC will not allow cells to enter anaphase until all the erroneous attachments are corrected except for the merotelic attachment that it does not sense. Therefore, cells can enter anaphase with merotelic chromosomes that were not corrected in the previous phase. Therefore, measuring the frequency of lagging chromosomes and the likelihood of forming micronuclei as suggested would not provide solid information concerning the functionality of the error correction mechanism.

9- Results, page 11, although a slight reduction in flux rates is somewhat indicative of alterations in microtubule dynamics, several other factors and processes (such as motor protein-mediated microtubule sliding) are involved in flux (see e.g. Steblyanko et al., EMBO J, 2020). A more direct measure of microtubule stability would be to calculate the half-life of the different spindle microtubule populations from the photoactivation experiments by fitting a double exponential curve to the fluorescence dissipation curves. Indeed, from the data provided, it seems that the photoactivation mark remains stronger over time after TLL11 depletion.

We agree with this reviewer about the spindle flux reflecting different processes, one of them being MT dynamics. As suggested we performed further experiments with the PA-tubulin cell line. Unfortunately, our microscope facility does not support the technical requirements needed to perform the analysis as described in the cited paper. We therefore could not obtain solid data on the turnover rate and half-life of kMT.

We nevertheless are confident that the data we provide on the change of spindle size, the increase of spindle MT cold sensitivity, the decrease velocity of the spindle flux, and the decrease in kinetochore distance are altogether convincing enough to support

our hypothesis of an increase in the stability of the spindle MTs in the TTLL11 silenced cells.

10- It remains unclear whether TTLL11 affects microtubule dynamics directly or indirectly. The authors briefly allude to this in the discussion, but this level of mechanistic insight would be expected to merit publication in Nature Communications. Obvious candidate proteins that are regulated by polyglutamylation, such as spastin (also implicated in spindle flux), should be tested, in addition to ruling out indirect effects over tubulin de-tyrosination and other well-established targets such as MCAK. We have now ruled out that other tubulin modifications such as tyrosination de-tyrosination or acetylation are altered in TTLL11 silenced cells (see new Fig.1c).

We are not aware of any data suggesting that tubulin polyglutamylation may impact directly MT stability. Instead there are strong evidences for MT polyglutamylation modulating the binding of MAPs as a function of the glutamate chain length. The data so far suggest that MT polyglutamylation could control the binding of specific MAPs to MTs in a length-dependent manner, without altering the binding of other MAPs. Some examples so far include structural MAPs such as tau, MAP2 and MAP1B as well as spastin and some kinesin motors (Hausrat et al, 2022, Lacroix et al, 2010; Ikegami et al, 2007, Bonnet et al, 2001).

We also monitored the localization of a few selected MAPs in control and siTTLL11 spindles. We did not detect any significant difference in localization of p150, Katanin, HKlp2 and Eg5 (see **figure 1** attached at the end for the reviewers). We also examined spastin and MCAK localizations. The two commercial antibodies we purchased for spastin (Santa Cruz - sc-53443 and Proteintech Ref. 227921-1-AP) did not give any convincing staining in mitotic cells. For monitoring MCAK we obtained a polyclonal antibody from Linda Wordeman. Again, we did not find any change in MCAK localization in siTTLL11 spindles (see **figure 2** attached at the end for the reviewers).

It is however possible that no major changes should be expected at the level of protein localization since bipolar spindles assemble and segregate chromosomes in the absence of TTLL11. Detecting changes in protein activity beyond their localization will require specific assays to be developed. Indeed, spastin for example has a higher MT severing activity on polyglutamylated MTs but its activity then decreases with longer E chains. This suggests a complex mechanism for the regulation of the binding and activity of associated proteins through MT polyglutamylation.

Polyglutamylation (as a general MT modification) will likely affect the interaction and binding kinetics of multiple proteins, making it unlikely that a single protein is responsible for mediating effects on the overall spindle dynamics and chromosome segregation fidelity. Further work that is out of the scope of the current manuscript will be needed, most probably using in vitro reconstitution approaches, to determine which motors and severing enzymes are sensitive to the degree of MT polyglutamylation and how this may account altogether for the mild changes in MT dynamics that we found in the spindles assembled in the absence of TTLL11.

Minor issues:

1- Introduction page 3: the statement about the SAC is incorrect, there are attachments that satisfy the SAC that are incorrect (merotelic). The origin of microtubules at kinetochores is also hotly debated and evidence that they (all) emanate from centrosomal microtubules is lacking. Sentence should be re-phrased.

This is indeed correct and we have revised this section of the introduction on page 3.

2- Top of page 4: it seems that there is a strikethrough in “the majority of”. Please correct.

Thank you for noticing this mistake. We have deleted it from the text.

Reviewer #3 (Remarks to the Author):

The authors show that TLL11 is responsible for polyglutamylation of the mitotic spindle in HeLa cells. To assess function of spindle polyglutamylation, they deplete TLL11 from HeLa cells and observe chromosome segregation defects. In parallel, they show that TLL11 depletion in zebrafish embryos leads to the formation of micronuclei. The authors then analyzed the TCGA database and found a systematic downregulation of TLL11 in cancer cells and they correlate this with aneuploidy. Finally, the authors show that spindle MTs are more stable in the absence of polyglutamylation. They conclude that polyglutamylation tunes MT dynamics to ensure faithful chromosome segregation.

Although understanding the role of spindle polyglutamylation during mitosis is a very interesting question, the data provided in this manuscript is not very convincing. The conclusions result from correlations between data obtained in different systems (HeLa cells, zebrafish embryos) and cancer cell databases without linking them together.

Major concerns:

Introduction Page 4: “During mitosis, spindle MTs are modified with numerous PTMs, including acetylation, detyrosination and polyglutamylation. However, still little is known about their roles in cell division, except for detyrosination, which guides metaphase chromosome congression via the motor protein CENP-E.”

While it is true that the function of polyglutamylation in mitosis remains unknown, a recent study highlights a role for MT acetylation in maintaining spindle bipolarity in epithelial cells (Rasamizafy et al. Cells 2021).

We apologize for this oversight, and we have now included this exciting new finding together with the reference in the introduction (page 4).

To explore whether TLL11 depletion may affect other tubulin PTMs we have quantified the levels of MT acetylation, tyrosination and detyrosination in spindles assembled in control and TLL11 silenced cells. We did not find any significant

changes in the levels of any of these PTMs (Fig.1c). Altogether, these data show that TTLL11 depletion specifically reduces spindle MT polyglutamylation levels without interfering with other PTMs. Therefore, our study shows that polyglutamylation by TTLL11 during mitosis represents a novel aspect of the role of a MT post-translational modification in mitosis that had not been addressed before.

Results Page 4: Among the different TTLL proteins, only TTLL11 and TTLL13 localize to the spindle when overexpressed. Wouldn't it be expected to have at least one of the initiating polyglutamylases? How do the authors explain this?

Both TTLL11 and TTLL13 were previously described as elongases (Van Dijk et al, 2007). However, TTLL11 was also found to have a low initiating activity when overexpressed in HeLa cells. One possibility is therefore that TTLL11 could provide both the initiation and elongation activities during mitosis.

Another alternative is that the initiating enzyme(s) interact only intermittently or very transiently with the microtubules, and cannot be solidly localised.

Finally, it is also possible that MTs primed during interphase by an initiating enzyme become good substrates for TTLL11 in mitosis. Indeed, interphase MTs have short branched chains of 1 to 2 glutamates labelled by the GT335 antibody (that recognizes the branching glutamate) but not the PolyE antibody (that recognizes chains of three or more glutamates) (Supp. Fig.2f,g). These are indeed good substrates for TTLL11 as previously shown when overexpressing the enzyme (Van Dijk, 2007).

Page 7: The authors show that zfTTLL11-MO injected embryos have severe developmental defects and conclude that this suggests "a pivotal role for TTLL11-dependent spindle MT polyglutamylation". To validate this conclusion, the authors should show that polyglutamylation is decreased or abolished on spindle MTs following TTLL11-MOs. Although polyglutamylation of spindle MT likely contributes to the observed phenotype, the authors cannot exclude a contribution of the polyglutamylation of other MTs, such as those from the nodal cilia known to play critical roles in early embryo patterning. Did the authors look whether the polyglutamylation levels in the zebrafish cilia are affected by TTLL11-MOs?

We now show that the levels of spindle MT polyglutamylation in dividing cells in zfTTLL11-MO-1 zebrafish embryos are significantly reduced compared to controls (Fig. 2f, g).

The chromosome segregation defects and micronuclei that we observed at early developmental times well before the start of embryo patterning fit well with those we obtained in HeLa and RPE1 cells (Fig1e, Supp. Fig.3c). They point to an essential role of TTLL11 and spindle MT polyglutamylation in chromosome segregation fidelity. Moreover, we found that the chromosome segregation defects and the developmental defects in TTLL11 morphant embryos can specifically be rescued by TTLL11 overexpression. Altogether, our data therefore suggest that the late zebrafish embryo

phenotypes observed at 36hpf may result in large part from the chromosome segregation defects occurring during the early embryonic divisions.

However as pointed out by this reviewer, we cannot exclude potential changes in MT polyglutamylation in the nodal cilia (or other MTs) that could contribute to the late embryonic phenotypes.

We have modified the sentence in the results section to:

'Altogether, our data suggested that TTLL11 plays an essential role during the early embryonic development phases.'

Page 7: The authors observe micronuclei in zebrafish embryos. Did they detect similar micronuclei in HeLa cells after TTLL11 siRNA treatment or did cells finally manage to divide properly? It would be interesting to know the consequences of the observed increase in lagging chromosomes in HeLa cells.

Indeed, it will be interesting to further investigate the consequences of a long-term silencing of TTLL11 in non-transformed and non-embryonic cells in the future.

HeLa cells are not the system of choice to look at the consequences of long-term TTLL11 silencing because they are aneuploid cancer cells that already have a basal rate of chromosomes mis-segregation and they have micronuclei.

We are currently setting up several tools to be able to address these questions in RPE1 cells in the future.

Page 9: Using TCGA database, the authors show a systematic downregulation of TTLL11 in cancer cells. Does this mean that spindle polyglutamylation is systematically downregulated in cancer cells? The authors use cancer cells (HeLa cells) for their study. Wouldn't it be more appropriate to use "normal" cells to address the role of spindle polyglutamylation? Do HeLa cells have low levels of polyglutamylation as compared to "normal" cells, for example how does their polyglutamylation level compare with the one of zebrafish embryo cells? The link between spindle polyglutamylation and cancer would be more convincing if authors had showed a systematic decrease in polyglutamylation in cancer cells.

We fully agree with this and have now quantified the levels of the spindle MT polyglutamylation in two colon cancer cell lines: HT-29, HCT116; two breast cancer cell lines: MDA-MB-231, MDA-MB-468 and the osteosarcoma cell line U2OS. We found that in all these cells the levels of the spindle MT polyglutamylation is very low in comparison with the levels we quantified in the non-transformed immortalized human hTERT-RPE1 cells, (Figure 5).

However, HeLa cells have relatively high levels of MT polyglutamylation in mitosis that are similar to those of RPE1 cells when compared to these other cancer cell lines (see **figure 3** attached at the end for the reviewers).

We acknowledge the importance to validate our data in normal cells and we have therefore examined now the consequences of silencing TTLL11 in the non-transformed hTERT-RPE1 cells. We found that TTLL11 silencing in hTERT-RPE1 also

reduces spindle MT polyglutamylation levels like in HeLa cells and in embryonic cells in vivo (Fig. 5 and Supp. Fig. 3a-b). We also show now that this results in an increase of the spindle length (Fig. 7b) and chromosome segregation defects (Supp. Fig. 3c).

Page 11: The authors found that siTLL11-depleted cells have more stable spindles. Do they observe changes in the levels of MT acetylation or detyrosination two PTMs known to increase with MT stability? Could these changes contribute to the observed phenotype?

This is indeed a very important point also raised by the other reviewers. We now addressed it directly by quantifying the levels of tyrosinated and detyrosinated MTs in spindles assembled in control and TLL11 silenced cells (Fig. 1c). We found no significant changes in the levels of spindle MT tyrosination or detyrosination in silenced cells. This suggests that the tubulin tyrosination-detyrosination cycle is not altered in TLL11 silenced cells.

In addition, we also checked whether for any change in MT acetylation levels in silenced cells. We did not find any difference between control and siTLL11 spindles (Fig.1c).

Overall, we conclude that TLL11 silencing specifically impairs MT polyglutamylation without interfering with other PTMs.

Minor points:

Top page 4: remove « the majority of ».
We have now removed these words.

Page 5: reference to figure S3b is misleading as it follows the sentence “ the polyE signal was specifically and significantly reduced in spindles assembled in siTLL11 cells as compared to control cells”. Figure S3b shows that polyE is not reduced in interphase cells depleted of TLL11. It might be more comprehensive for the readers to state this point more clearly. This would also help the readers to understand why in figure S2a, the basal polyglutamylation level of HeLa cells is not reduced after siTLL11 when examined by immunoblotting.

It was indeed not clear. We have revised this part of the result section to specifically state that the weak interphase PolyE signal is similar in control and siTLL11 cells both in IF and western blot analysis.

Movie S4: While H2A is labelled with mCherry and TLL11 with GFP, the colors are inverted in the movie. Same is true for the tubulin/H2B in Movie S3.

We thank the reviewer for pointing this out and we have now modified the colour code of the movies.

Top Page 11: TLL11 instead of TLL11 and K-fibres instead of K-fibers.

We have carefully checked the text throughout to conform to the English (UK). We used the term fibre throughout the text.

Reviewer #4 (Remarks to the Author):

Reviewer comments

In the manuscript, the authors identified that the enzyme TTLL11 regulates polyglutamylation and microtubule dynamics required for chromosomal stability during mitosis in vitro and in vivo. This enzyme polyglutamates the spindle microtubules in mitosis, which support microtubule dynamics and error-free chromosomal segregation. By using HeLa cell and zebrafish, the authors showed that decreasing TTLL11 expression level increased the probability of abnormal chromosomal segregation. Additionally, they found that TTLL11 is downregulated in human tumors, using a human cancer data base. Overall, the authors provide evidences for a new mechanism of spindle control by one specific post-translational modification of tubulin. Hence these results may be of interest to a broad audience. I included below major and minor points that will improve the quality of the manuscript and recommend them to be addressed prior to a possible publication.

Major comments

1. Expression of TTLL11

1-1. The authors overexpressed fluorescence tagged TTLLs as a strategy to visualize their localization. I understand that the absence of antibodies limit the analysis of the endogenous expression pattern. Nevertheless, I wonder if overexpressing TTLLs could affect their endogenous location due to for instance increased activity and tubulin glutamylation (as shown e.g. Fig S2A). One way to address this would be to provide the data for the localization and impact on polyglutamylation of the other overexpressed TTLLs described in the text, and/or to titrate the expression levels of GFP-TTLL.

We have now included in a supplemental figure the localizations of all the overexpressed fluorescently tagged TTLLs. It shows that TTLL11 and TTLL13 are the only two that when overexpressed localize to the spindle. Since only TTLL11 is expressed in HeLa cells we selected TTLL11 for further validation.

1-2. The GFP-TTLL11 expression in HeLa cells is observed in the entire cell in addition to the spindle (Figure 1A). On the other hand, the GFP-zTTLL11 expression in zebrafish blastomere seems to be more restricted in the spindle (Fig 2C). How this difference can be explained? Is it due to the overexpression strategy (e.g. different TTLL11 protein quantity)?

We fully acknowledge these comments. Indeed, the level of overexpression in HeLa cells is most probably higher than in zebrafish embryos. In HeLa cells, GFP-TTLL11

expression is driven by the CMV promoter which is commonly used for the production of high levels of recombinant protein in mammalian cells. We also analyzed localization shortly after transfection because we noticed that TTLL11 overexpression induced cell lethality. Overexpression of TTLL11 as well as that of other TTLLs has been shown to induce a massive polyglutamylation of the interphase MTs (VanDijk et al, 2007) that may explain the increase lethality.

Instead the expression of the zfTTLL11 in zebrafish embryos was driven by injection of mRNA encoding GFP-zfTTLL11 and visualized after several cycles of cell division. Since under these conditions we observed a rescue of the morphant embryo development, it is very likely that the levels of zfTTLL11 in these embryos is lower than in HeLa cells.

Overall it is likely that the differences we observed are indeed due to different levels of overexpression and the visualization methods (fixed versus live specimen).

2. Strategies to study the role of TTLL11

2-1. The authors used HeLa cells for the loss-of-function study of TTLL11. However, they also mentioned in the manuscript that the levels of TTLL11 is reduced in most tumors. I am wondering whether reducing TTLL11 in HeLa cells is the best approach to test the role of TTLL11 in microtubule dynamics and mitosis. I recommend the author to test their findings in other cells, if possible not cancerous cells.

We fully agree with this reviewer that it was important to address the role of TTLL11 in non-transformed cells. We have now further validated our data in the non-transformed immortalized hTERT-RPE1 human cells. We found that TTLL11 silencing also reduces spindle MT polyglutamylation levels (Fig.5, Supplementary Fig. 3a-b), promotes an increase of spindle length (Fig. 7b) and chromosome segregation defects (Suppl. Fig. 3c). These data are fully consistent with those we obtained in HeLa cells and in zebrafish embryos. Altogether they provide a solid support for the role of TTLL11 in regulating spindle polyglutamylation and chromosome segregation fidelity both in tissue culture cells and in vivo.

2-2. In the zebrafish model, the authors only used MO strategy to knockdown TTLL11. Even though they showed mRNA rescue experiment, the MO strategy is still arguable in the zebrafish society. Currently, F0 biallelic knockout by CRISPR/Cas9 is widely used in zebrafish. Hence, I suggest that they should show the identical phenotype via CRISPR/Cas9 system.

As suggested by the reviewer we have now performed a F0 biallelic zfTll11 knockout by CRISPR/CAS9.

The resulting phenotypes are fully consistent with those we obtained for morphant embryos (Supp. Fig.5):

- major embryonic developmental defects at 36hpf (Supp. Fig. 5a, b)
- a substantial increase of chromosome segregation defects including lagging chromosomes at earlier times of embryonic development (Supp. Fig. 5 c, d)

- an increase of embryos having cells containing micronuclei (Supp. Fig. 5e).

These results together with the rescue of the morphant phenotypes by expression of GFP-ZFTll11 showed that morpholino interference is specific for the downregulation of zFTLL11 in zebrafish embryos.

3. Analysis of knock-down phenotypes.

3-1. TTLL11 knock-down affect the levels of polyE tubulin in the spindle, but on the WB (Fig S2A), there is no effect on total polyE tubulin. In a similar line of thought, the authors do not show the impact of TTLL11 overexpression on polyE tubulin by staining. Given the WB shown in S2A, I would expect the increase of polyE tubulin to occur broadly in the cell. Taking these two observations into consideration, could the authors clarify how TTLL11 could specifically regulate the polyE on the spindle and not elsewhere?

TTLL11 silencing indeed does not change the basal PolyE signal detected by western blot as shown in Supp. Fig 2. Here we used non-synchronized cells that are therefore mostly in interphase. As previously described (Van Dijk et al, 2007), interphase MTs have a very low basal level of polyglutamylation as detected by IF (see Supp. Fig. 2d) and western blot analysis (see Supplementary Fig.2a). In the same blot (Supp. Fig.2a) we show that overexpression of GFP-TTLL11 causes an increase of interphase MT polyglutamylation, consistently with previous reports (Van Dijk et al, 2007). It means that when the enzyme is in the cytoplasm due to protein overexpression, it can polyglutamylate the interphase MTs.

GFP-TTLL11 localizes to the nucleus and the cytoplasm in interphase cells. This suggests that the endogenous protein that is less abundant is a nuclear protein. Indeed this fits well with the fact that the interphase MTs in non-transfected cells are not polyglutamylated. Our current hypothesis is therefore that TTLL11 may be a novel RanGTP regulated protein that specifically functions around the chromosomes during mitosis and meiosis like other previously characterized SAFs (Spindle Assembly Factors) (Cavazza et al, 2016). We plan to directly address this hypothesis in the future.

3-2. The authors showed abnormal chromosome segregation and severe developmental defects in zebrafish by downregulating TTLL11. It seems that the abnormal segregation generates aneuploidy and chromosome instability. However, the authors did not show that directly. The authors should provide a direct evidence that TTLL11 downregulation induces aneuploidy.

We have now included additional data on TTLL11 interference in the non-transformed human cell line hTERT-RPE1 cells, and using a bi-allelic TTLL11 knockout strategy in zebrafish embryos. In all these conditions TTLL11 reduction results in an increase of chromosome segregation defects in anaphase (Fig. 1e, 3b, Supp. Fig. 3c, Supp. Fig. 5d) and the presence of micronuclei (Fig. 3 c, d, e and Supp. Fig. 5e).

We also traced the origin of several micronuclei in zebrafish embryos to lagging chromosomes during anaphase (Fig. 3c).

We believe that altogether these data can convincingly show that TTLL11 downregulation does indeed promote aneuploidy.

3-3. In the current manuscript, the authors did not explain sufficiently why the zebrafish have such strong developmental defects. Are these due to e.g., increased apoptosis or reduced proliferation? It is particularly interesting that zebrafish show such a severe phenotype given that HeLa cells do not show any changes in their mitotic index. I would recommend the authors to describe and discuss better how similar subcellular defects (genome instability) in the zebrafish embryos and HeLa cell line show different phenotypes (proliferation vs developmental defect/apoptosis).

We have performed additional experiments in control and morphant (MO) zebrafish embryos to directly address any potential effect of TTLL11 downregulation on promoting apoptosis and/or reducing cell proliferation during early embryonic development.

We did not find any evidence for an increased rate of apoptosis in MO embryos as assessed by Caspase 3 antibody staining (see **figure 4** attached at the end for the reviewers).

To assess a potential effect on proliferation we measured the cell size in control and MO embryos at blastula/gastrula stage (4 hpf). In principle, a decreased mitotic rate should result in a larger cell size in embryonic progenitor cells after the cleavage period. We did not find any significant difference in cell size between control and MO embryos. We therefore conclude that the cell proliferation rate is not affected by Tll11 downregulation (see **figure 5** attached at the end for the reviewers).

Overall, our data directly show that chromosome segregation fidelity is compromised in the MO and bi-allelic knockout embryos during the early developmental phases that mainly consist of rapid cell divisions. These errors may have a major contribution to the strong developmental effects we observed at later development times. However, we cannot rule out some additional effects during the developmental stages following these initial cell divisions that we plan to investigate in the future.

It is quite difficult to compare the data from zebrafish embryos with those of HeLa cells. HeLa cells are immortal human cells isolated many years ago from a cervical carcinoma. They are highly aneuploid with a basal rate of chromosome mis-segregation events and micronuclei. We did not follow cells under sustained downregulation of TTLL11 (like we did in zebrafish embryos) but rather analyzed the first mitotic events after TTLL11 acute downregulation.

By contrast, zebrafish embryo cells are normal cells and the developmental phenotypes we observed were recorded after several rounds of cell division under constant downregulation (MO) or elimination (biallelic knockout) of Tll11 expression.

Overall, however, the observed phenotypes are consistent: HeLa cells show an increase of chromosome segregation errors upon acute elimination (or reduction) of TTLL11 (and we provide now similar data for the untransformed human cell line RPE1) and zebrafish embryos show increased rates of chromosome segregation errors and micronucleated cells during the early phases of development under sustained elimination/reduction of Tll11 which is associated with strong developmental defects at later stages.

3-4. Currently the authors do not show if zebrafish embryos injected with the TTLL11 MO show reduced polyE glutamylation in the spindle. Adding these results is crucial to further validate that the observations in downregulated HeLa cells hold true in the zebrafish too.

We thank the reviewer for pointing out this very important issue. We now provide the quantification of the levels of spindle MT polyglutamylation in control and MO embryos. We found that indeed polyE levels are significantly reduced in these embryos (Fig.2f-g). Moreover, in agreement with our data in HeLa cells (Fig.7a) and in RPE1 cells (Fig. 7b), we also found that in the absence of TTLL11 the spindles are longer than in control embryos (Fig.7c).

4. Additional discussion

The authors mainly focused the localization and function of TTLL11 in mitosis. However, they still emphasize polyglutamylation in the chromosomal fidelity in the manuscript. I noticed that TTLL13 is also localized in the spindle at mitosis, and it is increased in several tumors in contrast to TTLL11. It might be worthy to describe about this additional player for glutamylation in the discussion part.

We decided to focus on TTLL11 because in contrast to TTLL13, it is widely expressed in human tissues and in HeLa cells.

TTLL13 is barely expressed in most human tissues and it is also not expressed in HeLa cells (Supplemental Fig.1c). It was described as an elongase. We do not know the functional implications of its overexpression in cancer but it does not fit with the reduced polyglutamylation of the spindle MTs, suggesting that its function may be related to another process. Moreover, although TTLL13 is overexpressed in human cancer, its levels still remain very low and in fact lower than those of downregulated TTLL11 (Supp. Fig. 6g).

Our new studies now provide additional data that support the unique role of TTLL11 amongst the TTLL glutamylases in cancer (Fig. 6; Suppl. Fig. 8a-c).

In addition, we show now that in agreement with the downregulation of TTLL11 in cancer, the levels of spindle MT polyglutamylation are reduced in cancer cell lines (Fig.5).

Minor comments

1. The authors did not describe what TTLL11 stands for. Please provide the full name before using abbreviation.

As suggested, we have now added the full name of the enzyme (Tubulin Tyrosine Ligase Like 11) after it is first mentioned in the introduction (page 4).

2. The supplementary movie files are not in order. For example, the authors described the movie S5 before using the movie S2. Please arrange the name of files in orders for readers' convenience.

We have carefully checked the order and reorganized the files so that they are now cited in the right order within the text.

3. There are a few typos, abnormal marks or reference not added. Please fix them.

- Page 4, line 1: the majority of
- Page 11, line 2: TTLL111
- Page 11, line 7: (Warren et al, 2020).
- Page 32, line 1: .p, 0.01

We have carefully revised the text and corrected all these errors.

4. Please avoid the use of red and green colors in your merge confocal image for color-blind accessibility.

We have checked the figures for any data for which visualizing distinct localizations is essential and requires the visualization of the two colours. We believe that the only image that indeed needed to be adapted for colour blind people is the image shown in previous Fig.5c (now Fig. 7e). We have therefore modified this image using a magenta, yellow and blue combination.

Figure 1

Metaphase localization of various MT associated proteins in control and siTTLL11 HeLa cells

There are no significant differences in the localizations of a component of the dynactin complex p150, the severing enzyme katanin and the motor proteins hKlp2 and Eg5.

(A) Immunofluorescence images of metaphase spindles in control and siTTLL11 cells stained for p150. The bar plot shows quantification of the p150 signal normalized on the total tubulin signal. n (control) = 41 cells and n (siTTLL11) = 42 cells. **(B)** Immunofluorescence images of metaphase spindles in control and siTTLL11 cells stained to visualize Katanin. The graph shows the quantification of the Katanin signal normalized on the total tubulin signal. n (control) = 66 cells and n (siTTLL11) = 49 cells. **(C)** Immunofluorescence images of metaphase spindles in control and siTTLL11 cells stained to visualize HKLP2. The graph shows the quantification of the HKLP2 signal normalized on the total tubulin signal. n (control) = 45 cells and n (siTTLL11) = 51 cells. **(D)** Immunofluorescence images of metaphase spindles in control and siTTLL11 cells stained to visualize Eg5. The graph shows the quantification of the Eg5 signal normalized on the total tubulin signal. n (control) = 41 cells and n (siTTLL11) = 43 cells.

Scale bars, 10 μm . Error bars represent SD. Tubulin (red), protein of interest (green) and DAPI (blue). p -values are based on t-test with a 95% confidence interval.

Method

HeLa cells were grown on glass coverslip and fixed in ice-cold methanol for 3min at -20C° . The following primary antibody were used: mouse anti- α -tubulin (DM1A, Sigma T9026) diluted 1:1000; rabbit anti-p150 (BD Bioscience, 610474), diluted 1:750; rabbit anti-Eg5 (BD Bioscience, 61186) diluted 1:1000; rabbit anti-katanin (KATNB1, Proteintech, 14969-1AP) diluted 1:500; rabbit anti-Hklp2 (home-made, 0,8 $\mu\text{g}/\mu\text{l}$). DNA was counterstained with DAPI (1 $\mu\text{g}/\text{ml}$; Sigma-Aldrich) diluted 1:1000. Antibodies were diluted in the following buffer: PBS 1x, 0,1% Triton-X-100 (v/v), 0,5% BSA (w/v) Images were acquired with Leica DEI 6000B microscope mounted with a DF2 9000GT camera and processed with ImageJ Fiji software. Quantifications were performed as described in the main text of the manuscript.

Figure 2

MCAK localization in control and siTTLL11 HeLa cells

There are no significant differences in MCAK localization in control and siTTLL11 mitotic cells.

Immunofluorescence images of control and siTTLL11 HeLa cells in prophase (upper panels) and metaphase (lower panels). The red arrows point to MCAK localization to the kinetochores in prophase cells. Tubulin is shown in green, MCAK in red and DNA in blue. Scale bar, 5 μ m

Method

HeLa cells were grown on glass coverslip and fixed in ice-cold methanol for 3 min at -20 C°. Antibodies were diluted in the following buffer: 1' PBS, 0.1% Triton-X-100 (v/v), 0.5% BSA (w/v). Primary antibodies were mouse anti- α -tubulin (DM1A, Sigma T9026), diluted 1:1000 and sheep anti-MCAK (generous gift from Linda Wordeman), diluted 1:100. DNA was counterstained with Hoechst 33342, at 1:1000. Images were acquired with a LeicaDMI 6000B microscope, 100x objective, mounted with a DF2 9000GT camera, and processed using ImageJ Fiji software.

Figure 3

Spindle MT polyglutamylation in HeLa and RPE1 cells

The normalized levels of spindle MTs polyglutamylation detected with the PolyE antibody are similar in HeLa and RPE1 cells.

(A) Immunofluorescence images of control and TTLL11 HeLa and RPE1 cells.

In the merge, tubulin is in red, polyE is in green and DNA in blue. Scale bar, 10µm.

(B) Quantification of the polyE signal normalized to the total tubulin signal in control and siTTLL11 spindles in HeLa and RPE1 cells; $n_{HeLa}(\text{control}) = 10$ cells; $n_{HeLa}(\text{siTTLL11}) = 12$ cells; $n_{RPE1}(\text{control}) = 8$ cells and $n_{RPE1}(\text{siTTLL11}) = 19$ cells. Graph corresponding to one experiment. Error bars represent SD. p-values: (****) ≤ 0.0001 , (*) $\leq 0,05$, based on one-way ANOVA test with a 95% confidence interval.

Method

As described in the Method section of the main manuscript

Figure 4

Cell apoptosis in morphant zebrafish embryos

There are no significant differences in apoptotic cell levels in control and morpholino injected embryos at 6hpf and 8hpf. The number of embryos showing apoptotic cells is also similar at both stages in controls and morphants.

(A) Representative images of cells obtained from control (left) and Ttl11 MO embryos (right) at two different stages (6 hpf and 8 hpf) (B) Quantification of apoptotic cell levels in control (CTRL) and Ttl11 MO injected embryos (MO inj) at 2 different stages (6 hpf and 8 hpf) (n=30 embryos per condition).

Method

To measure cell apoptosis, 2.8 ng of Ttl11 morpholino were injected into one-cell stage wildtype embryos. 30 wild type and morpholino injected embryos were fixed with 4% of PFA at 6 hpf and 8 hpf. Then, a whole-mount immunofluorescence was performed to detect apoptosis, following the protocol published by Sorrells et al. 2013. A Casp3 antibody (BD Biosciences, 15889738) was used at 1:250 to visualize active caspase 3 through a secondary antibody (anti-rabbit Alexa 488). Pictures were taken in an Olympus scope (SZx16) to quantify apoptotic cell levels using ImageJ.

References:

Sorrells, S., Toruno, C., Stewart, R.A., Jette, C. Analysis of Apoptosis in Zebrafish Embryos by Whole-mount Immunofluorescence to Detect Activated Caspase 3. *J. Vis. Exp.* (82), e51060, doi:10.3791/51060 (2013).

Figure 5

Cell size in morphant zebrafish embryos

There are no significant differences in cell size in control (ctrl-green) and TLL11 morpholino injected (MO inj-yellow) embryos at the same developmental stage.

(A)

(B)

(A) Representative brightfield images of cells obtained from control (left) and Tll11 MO embryos (right). (B) Analysis of cell sizes obtained in three independent experiments comparing the cell diameter of controls versus morpholino injected embryos (n=30 cells for each condition).

Method

To measure cell size, 2.8 ng of Tll11 morpholino were injected into one-cell stage wildtype embryos. 20 embryos were manually dechorionated in E3 buffer at 3.5-4 hpf. The embryos were transferred to an Eppendorf tube containing 500 µl of D3 culture medium (D2906, Sigma) and were mechanically dissociated by manual tapping followed by centrifugation at 200 g for 3 min. After one wash, single cells were plated in a Mattek dish and visualized through a Leica DM IL LED microscope using a 10x objective. Cell size measurements of 30 individual cells per condition were taken using *ImageJ* and the cell diameter was calculated as the average between the major and minor axis of each cell.

REVIEWERS' COMMENTS

Reviewer #1 (Remarks to the Author):

The authors have added a large number of new data to address my questions and solidify their conclusion. I am glad to recommend the revised paper for publication.

Minor point:

In the point-by-point response, authors claimed that TTLL11-YFP was expressed in Supplementary Fig 4C. However, the label in corresponding figure is inconsistent (TTLL11-GFP).

Reviewer #2 (Remarks to the Author):

The authors have satisfactorily addressed all my concerns and did a great job overall in revising all the main points raised by the four reviewers. I am therefore happy to recommend this work for publication in Nature Comms.

Reviewer #3 (Remarks to the Author):

The authors have greatly improved their manuscript and now provide stronger evidence for a role of TTLL11 in faithful chromosome segregation. It is disappointing however that the molecular mechanisms implicating microtubule polyglutamylation in chromosome segregation are still poorly explained.

Minor points :

Lane 180-181 : remove « that ».

Reviewer #4 (Remarks to the Author):

The authors have performed many new experiments to address the reviewers' comments. Overall their findings and conclusions are now consolidated by their new experiments. I do not have any further comments.

Answers to the reviewer's comments

Reviewer #1

In the point-by-point response, authors claimed that TTLL11-YFP was expressed in Supplementary Fig 4C. However, the label in corresponding figure is inconsistent (TTLL11-GFP).

We apologize for this mistake. The label of the figure corresponds to the correct transfected constructs.

In our response, we had meant to say:

'We transfected HEK293 cells with constructs expressing GFP-TTLL11 or **TTLL11-GFP** or TTLL11-YFP-E531G. In agreement with previous reports, Western blot analysis of the cell lysates showed that TTLL11 overexpression drives a substantial increase of tubulin polyglutamylation (Van Dijk et al 2007).'

Reviewer #3

Lane 180-181 : remove « that ».

We removed the word "that" from line 180.